# Transcriptome Analysis of Muscle Tissue from Three Anatomical Locations in Male and Female Kazakh Horses

**DOI:** 10.3390/biology14091216

**Published:** 2025-09-08

**Authors:** Ayixie Wubuli, Yi Su, Xinkui Yao, Jun Meng, Jianwen Wang, Yaqi Zeng, Linling Li, Wanlu Ren

**Affiliations:** 1College of Animal Science, Xinjiang Agricultural University, Urumqi 830052, China; m15199854723@163.com (A.W.); 13335339131@163.com (Y.S.); yaoxinkui@xjau.edu.cn (X.Y.); junm86@xjau.edu.cn (J.M.); dkwjw@xjau.edu.cn (J.W.); xjauzengyaqi@163.com (Y.Z.); lilinling@xjau.edu.cn (L.L.); 2Xinjiang Key Laboratory of Equine Breeding and Exercise Physiology, Urumqi 830052, China

**Keywords:** Kazakh horse, skeletal muscle, RNA-seq, differentially expressed gene

## Abstract

This study performed comparative transcriptomic analyses of the longissimus dorsi, rectus abdominis, and diaphragm muscles in eight Kazakh horses (four stallions and four mares). Several differentially expressed genes (DEGs) were identified between sexes. Gene Ontology (GO) and the Kyoto Encyclopedia of Genes and Genomes (KEGG) pathway analyses revealed pronounced sexual dimorphism in muscle gene expression and structural organization. Notably, cytoskeleton-associated genes, including *TPM1*, *MYL1*, *MYH3*, and *PYGM*, emerged as potential key regulators mediating sex-specific muscle development. The results provide molecular evidence for elucidating the mechanisms underlying sexual dimorphism in equine skeletal muscle growth.

## 1. Introduction

Kazakh horses, one of the most representative traditional livestock breeds, possess skeletal muscle characteristics closely linked to their athletic performance, such as endurance for long-distance running and load-bearing capacity. These traits constitute critical targets for breed selection and functional optimization [1,2]. Skeletal muscle is a principal component of the mammalian locomotor system. Its development involves mechanisms such as muscle fiber differentiation, metabolic regulation, and regeneration, all of which directly influence athletic performance [3,4,5]. Beyond locomotion, skeletal muscle also plays a pivotal role in overall physiological function and metabolic health. Strength training can improve maximum voluntary contraction (MVC) activity, while low-intensity resistance exercises can increase the oxygen consumption of skeletal muscles [6,7]. In equines, athletic capacity is fundamentally determined by the structure and function of skeletal muscles, which are regulated by muscle fiber types, metabolic mechanisms, genetic background, postnatal training, and nutritional management [8]. As the driving force of locomotion, the structure and function of skeletal muscles ultimately determine performance metrics such as speed, endurance, and explosive power during physical exertion [9].

Energy supply in skeletal muscle operates through two primary metabolic pathways: aerobic and anaerobic metabolism. Aerobic metabolism relies on mitochondrial oxidation of fatty acids and glycogen to generate abundant ATP with minimal lactate accumulation, thereby supporting prolonged exercise. In contrast, anaerobic metabolism rapidly generates energy through glycolysis, which results in lactate accumulation and leads to muscle acidification and fatigue [10].

This study focused on three functionally distinct muscles: the longissimus dorsi, the rectus abdominis, and the diaphragm. The latissimus dorsi comprises a greater proportion of slow-twitch muscle fibers, which are well suited for oxidative metabolism and maintain muscle endurance [11,12]. The rectus abdominis contributes to core stability and affects muscle explosive power through fast-twitch muscle fibers [4,13]. The diaphragm, as the primary respiratory muscle, directly regulates oxygen intake and influences aerobic performance [14].

Sexual dimorphism affects athletic performance through mechanisms such as hormone regulation, metabolic pathways, and muscle structure remodeling [15]. Androgens promote muscle protein synthesis by activating the mTOR signaling pathway, thereby influencing muscle fiber composition and enhancing athletic performance in males [16,17]. Androgen receptors (ARs) in mesenchymal stem cells regulate the expression of IGF1, which promotes skeletal muscle protein synthesis and improves male athletic performance [18]. In contrast, estrogens in females enhance mitochondrial biogenesis and fatty acid oxidation efficiency through the AMPK signaling pathway, thereby improving aerobic endurance [19]. During post-exercise recovery, estrogen enhances antioxidant and anti-inflammatory responses by regulating *BDNF* expression, thus accelerating muscle repair [20].

Existing studies primarily focused on age-related changes and region-specific differences in the skeletal muscles of Kazakh horses of the same sex [1]. However, few involved gender differences at the transcriptional level in muscle sites. In this study, although direct performance traits such as oxygen consumption and lactate threshold were not measured, RNA sequencing (RNA-seq) was employed to compare sex-specific gene expression profiles across three functionally distinct muscle groups: longissimus dorsi, rectus abdominis, and diaphragm muscles. By integrating molecular data with quantitative histology and metabolite composition analyses, these findings provide molecular insights into the mechanisms underlying sex differences in equine skeletal muscle growth.

## 2. Materials and Methods

### 2.1. Experimental Animals and Sample Collection

Eight clinically healthy Kazakh horses, consisting of four three-year-old stallions and four three-year-old mares, were selected from Tacheng, Xinjiang, China. All horses were raised under the same feeding conditions, provided with high-quality alfalfa hay and corn kernels, and had free access to water. Immediately after slaughter, muscle specimens were collected from each stallion and mare as follows: longissimus dorsi, 3 cm lateral to the transverse process of the 13th thoracic vertebra; rectus abdominis, 3 cm lateral to the external sheath of the linea alba; and diaphragm, at the rib attachment site near the costal angle. Full-thickness muscle samples with a depth of 3 cm were synchronously obtained. Each group comprised four biological replicates, with individual sample weights standardized at 30 g.

Figure 1 illustrates the experimental procedure: a portion of the samples was immediately flash-frozen in liquid nitrogen (−196 °C) for subsequent RNA sequencing and biochemical analyses (fatty acid and amino acid content determination), while another portion was quickly fixed in 4% paraformaldehyde (PAF) for paraffin section preparation.

### 2.2. Histochemical Analysis

Muscle samples fixed in paraformaldehyde were subjected to dehydration using a gradient of ethanol (75%, 85%, 90%, 95%, and 100%). Samples were then cleared with xylene, embedded in paraffin, and sectioned into 5 μm cross-sections using a rotary microtome. After being stained with hematoxylin and eosin (H&E), the tissue slices were photographed using a light microscope (Eclipse E100 Nikon, Nikon Corporation, Tokyo, Japan) connected to a camera system.

### 2.3. Determination of Fatty Acid and Amino Acid Content

For fatty acid analysis, approximately 50 mg of the sample was homogenized. Then, 3 mL of n-hexane was added, and the mixture was shaken at 50 °C for 30 min. Subsequently, 3 mL of methanolic KOH solution (0.4 mol/L) was added and shaken at 50 °C for another 30 min for derivatization. Upon standing and cooling to room temperature, 1 mL of water was added and mixed thoroughly. The supernatant was collected after standing and diluted fivefold. From the diluted solution, 90 μL was taken, to which 10 μL of internal standard (methyl nonadecanoate, 125 μg/mL) was added, followed by analysis using gas chromatography-mass spectrometry (GC-MS, Agilent 7890B-5977B, Agilent Technologies, Santa Clara, CA, USA).

For amino acid analysis, 2 mL of 6 mol/L hydrochloric acid was added to the sample under nitrogen protection. Acid hydrolysis was performed at 110 °C for 24 h. Then, 100 μL of the hydrolysate was taken and evaporated to dryness at 40 °C under nitrogen using a nitrogen blower. The residue was reconstituted with 1 mL of water. For both mixed standards and test samples, 50 μL of solution was mixed with 50 μL of protein precipitant (10% sulfosalicylic acid containing NVL), vortexed, and centrifuged at 13,200 rpm under cooling conditions for 4 min. Next, 8 μL of the supernatant was mixed with 42 μL of borate buffer (pH 8.5), vortexed, and briefly centrifuged. Then, 20 μL of AQC derivatization reagent was added, vortexed, briefly centrifuged, and incubated at 55 °C for 15 min for derivatization. The resulting sample was cooled in a refrigerator, mixed thoroughly, and centrifuged again. A 50 µL of the supernatant was analyzed using an ultra-high-performance liquid chromatography-quadrupole ion trap tandem mass spectrometer (UHPLC-MS/MS, Waters ACQUITY UPLC I-Class/Xevo TQ-S, Waters Corp., Milford, MA, USA).

Quantitative data for fatty acids and amino acids were initially processed using Microsoft Excel. Independent-sample *t*-tests were then performed in SPSS 19.0. Data are presented as mean ± standard deviation (mean ± SD). Differences were considered highly significant at *p* < 0.01 and significant at *p* < 0.05.

### 2.4. Transcriptomic Sequencing

RNAs were extracted from the longissimus dorsi, rectus abdominis, and diaphragm tissues. After quality control, high-quality RNAs were purified, fragmented, and reverse-transcribed into complementary DNAs (cDNAs) to construct transcriptomic libraries. Sequencing was performed by Hangzhou Repugene Technology Co., Ltd. on the Illumina NovaSeq6000 platform (Repugene Technology, Hangzhou, China) [21].

### 2.5. Bioinformatics Analysis

FastQC (fastqc_v0.11.8) was employed to assess the quality metrics of raw Illumina sequencing data. Adapter sequences and low-quality reads were filtered using fastp (fastp 0.23.1), yielding high-quality clean reads. Bismark (version 0.24.0) was subsequently applied to map the clean reads to the Equus Caballus reference genome (EquCab3.0). Reads successfully aligned to the reference genome were designated as target sequences for subsequent standardized and customized analyses [22].

Filtering parameters included the elimination of reads containing adapter contamination, those with terminal base quality scores below 3, or ambiguous bases (N). A sliding window approach, with a four-base window and a quality threshold of 15, was applied to truncate reads once the average quality within the window dropped below the threshold. Reads shorter than 36 nt post-trimming or lacking valid pairing were excluded from further analysis.

### 2.6. GO and KEGG Enrichment Analyses

In order to annotate the differentially expressed mRNAs, GO and KEGG enrichment analyses were performed. KOBAS software (version 3.0) was used to assess the statistical enrichment of DEGs in KEGG pathways, while GOseq software (version 1.56.0) was adopted for GO functional analysis. Statistical significance for enrichment was set at *p* < 0.05.

### 2.7. Protein–Protein Interaction (PPI) Network Analysis

Based on the intersection of DEGs and known protein interaction pairs retrieved from the STRING database (https://cn.string-db.org, accessed on 25 July 2025), a PPI network was constructed. Homologous protein interaction relationships were integrated into the network. Cytoscape (version 3.10.0) was employed for network visualization and analysis. Specifically, the “Betweenness-unDir” plugin within the CytoNCA module was utilized to identify key targets and visually analyze the results.

### 2.8. RT-qPCR Validation

In order to validate the expression of individual mRNAs, total RNA was reverse-transcribed into cDNA. RT-qPCR primer information is provided in Appendix A. RT-qPCR was conducted on a CFX Connect Fluorescence Quantitative PCR system (Bio-Rad Laboratories, Inc., Hercules, CA, USA) with three replicates per sample.

### 2.9. Ethical Statement

The experimental protocol and procedures were approved by the Animal Ethics Review Committee of Xinjiang Agricultural University (Approval No. 2023004).

## 3. Results

### 3.1. Histochemical Characteristics of Muscle Fibers

H&E staining revealed red-stained cytoplasm with clear cell boundaries, while nuclei were stained blue. Each individual muscle fiber contained multiple nuclei arranged regularly along the sarcolemma edges (Figure 2). This study evaluated regional muscle development status by measuring muscle fiber diameter, density, and cross-sectional area. As shown in Figure 3, compared to the stallion horse diaphragm group (Mg), the mare horse diaphragm group (Gg) exhibited a significantly larger average muscle fiber cross-sectional area (*p* < 0.05) and lower muscle fiber density (*p* < 0.05). No significant differences were observed in muscle fiber diameter between groups (*p* > 0.05).

### 3.2. Comparative Analysis of Fatty Acid and Amino Acid Compositions

To assess sex-related differences in fatty acid and amino acid composition between Kazakh stallions and mares, the contents of fatty acids and amino acids were measured in the longissimus dorsi, rectus abdominis, and diaphragmatic muscles. As shown in Table 1, stallions exhibited significantly higher levels of C18:3n3 and C20:3n3 in the longissimus dorsi compared to mares (*p* < 0.01). In the rectus abdominis muscle, stallions displayed markedly higher levels of C18:2n6c and C20:2 (*p* < 0.01), as well as significantly elevated C24:0 (*p* < 0.05). No significant sex differences were observed in the fatty acid content of the diaphragm (*p*> 0.05). Regarding amino acids (Table 2), no evident differences were detected in the longissimus dorsi (*p* > 0.05). However, serine and cysteine levels in the rectus abdominis muscle were significantly higher in stallions than in mares (*p* < 0.01), and glycine content in the diaphragm was markedly elevated in stallions compared to mares (*p* < 0.01).

### 3.3. Sequencing Quality Analysis

A total of 12 transcriptomic samples were collected from the longissimus dorsi, rectus abdominis, and diaphragm of stallion and mare Kazakh horses. The quality control results are shown in Table 3. A total of 1,265,907,346 clean reads were obtained (with an average of 52,746,139.42 clean reads per sample). The error rate was 0.02%, with Q20 over 97% and Q30 exceeding 93%. The GC content ranged from 51.09% to 53.09%. Both the Kazakh stallion (G) and Kazakh mare (M) groups showed a passing rate greater than 90%. These results indicate that the transcriptome sequencing data are of high quality and appropriate for subsequent analysis.

### 3.4. Sample Correlation Analysis

For mares, transcript abundance was highest in the diaphragm muscle group (Mg) and lowest in the longissimus dorsi muscle group (Mb) (Figure 4A). Expression differences among individual samples were minimal, and the expression trends were largely consistent across individual samples. Furthermore, Figure 4B demonstrates a high degree of correlation among samples within each group.

### 3.5. DEG Screening

Differential expression analysis revealed that, compared to Gb, Mb exhibited 214 upregulated and 147 downregulated genes (Figure 5A), including *CHKB*, *WIPI1*, *DMPK*, and *ATP2A2* (Table 4). Mf showed 131 upregulated and 99 downregulated genes relative to Gf (Figure 5B), such as *HSPB2*, *CSNK1D*, *PROX1*, and *GSS* (Table 5). Comparatively, in Mg versus Gg, 131 genes were upregulated and 105 were downregulated in Mg (Figure 5C), including *OSCP1*, *UCKL1*, *MTMR7*, and *ANGPTL4* (Table 6). All the DEGs are detailed in Appendix A.

Cluster analysis, as presented in Figure 6A–C, indicated that the DEGs in the longissimus dorsi, rectus abdominis, and diaphragm tissues of male and female Kazakh horses were highly reproducible, suggesting significant differences between the groups.

### 3.6. GO Functional Annotation and KEGG Enrichment Analysis

For the Mb and Gb groups, as shown in Figure 7A, the DEGs were primarily enriched in biological process (BP) terms, including muscle system process, muscle structure development, muscle cell differentiation, striated muscle cell differentiation, and muscle cell development; cellular component (CC) terms including contractile fiber, longitudinal sarcoplasmic reticulum, sarcomere, myofibril, and myosin II complex; molecular function (MF) terms such as nucleotide binding, adenyl nucleotide binding, purine nucleotide binding, and small molecule binding. As illustrated in Figure 7B, KEGG analysis revealed significant enrichment of DEGs, including *TPM1*, *MYL1*, and *MYH3*, in pathways such as cytoskeleton in muscle cells, alanine aspartate and glutamate metabolism, cGMP-PKG signaling pathway, central carbon metabolism in cancer, and Th17 cell differentiation.

Regarding the Mf and Gf groups, as shown in Figure 8A, DEGs were enriched in BP terms, including muscle system process, nucleoside monophosphate metabolic process, regulation of protein catabolic process, and organophosphate metabolic process; CC terms, such as contractile fiber, sarcomere, supramolecular fiber, and myosin complex; and MF terms including natriuretic peptide receptor activity, nucleotide binding, purine nucleotide binding, ribonucleoside triphosphate phosphatase activity, and ATP hydrolysis activity. KEGG analysis results in Figure 8B indicated that the DEGs were mainly enriched in metabolic pathways, the cytoskeleton in muscle cells, the glucagon signaling pathway, the AMPK signaling pathway, and the mTOR signaling pathway.

As for the Mg and Gg groups, GO annotation (Figure 9A) showed that DEGs were primarily enriched in BP terms such as muscle filament sliding, actin-myosin filament sliding, actin-mediated cell contraction, striated muscle contraction, and muscle system process; CC terms including myofibril, sarcomere, contractile fiber, muscle myosin complex, and myosin II complex; and MF terms such as structural constituent of muscle, protein binding, actin monomer binding, myosin phosphatase activity, and troponin T binding. KEGG pathway analysis, as shown in Figure 9B, highlighted significant enrichment in pathways such as the cytoskeleton in muscle cells, hypertrophic cardiomyopathy, motor proteins, regulation of actin cytoskeleton, and cardiac muscle contraction (Appendix A).

### 3.7. PPI Network Analysis

PPI networks were constructed based on DEGs from three anatomical regions of male and female Kazakh horses. In the Mb versus Gb comparison, core genes identified included *RACK1*, *PYGM*, *MRPL4* and *PKM*; in the Mf vs. Gf group, *TICAM1*, *CLFAR*, *ITCH* and *PCMT1* were highlighted; and in the Mg vs. Gg group, core genes such as *MYH3*, *MYH2*, *MYH8*, *MYL2* and *MYL3* were identified (Figure 10).

### 3.8. RT-qPCR Analysis

In order to validate the reliability of the DEG data obtained from RNA-Seq, seven DEGs from the longissimus dorsi muscle were randomly selected for RT-qPCR analysis. The expression levels of these genes were consistent with the RNA-Seq results, exhibiting similar upregulation or downregulation trends, confirming the data reliability (Figure 11).

## 4. Discussion

Skeletal muscle development in horses is regulated by multiple factors, including age and sex. Ren et al. [1] investigated skeletal muscle development in Kazakh horses of varying ages and identified *BMP2*, *MDH1*, and *ATF3* as key genes involved in this process. Wang et al. [2] found that *RYR3* and *MYH6* genes regulate fast and slow muscle function in stallion and mare Kazakh horses, potentially improving meat quality by altering muscle fiber composition. Despite these insights, studies specifically examining the effect of sex on equine skeletal muscle development remain limited [8]. Therefore, this study selected eight three-year-old Kazakh horses (four stallions and four mares) and performed transcriptome sequencing analysis and histochemical analysis on samples from their longissimus dorsi, rectus abdominis, and diaphragm.

KEGG enrichment analysis revealed that DEGs were significantly enriched in pathways such as Cytoskeleton in muscle cells, the thyroid hormone signaling pathway, and the regulation of actin cytoskeleton. The cytoskeleton, primarily composed of microtubules, actin filaments, and intermediate filaments, plays an essential role in multiple fundamental cellular and BPs, such as cell migration, movement, division, and the establishment and maintenance of cellular and tissue structures [23]. It forms a dynamic yet mechanically stable network of cytoskeletal filaments. Muscle contractile force is generated by actin and myosin filaments, among which the orderly arrangement of polar actin filaments is particularly crucial for striated muscle contraction [24].

Jabre et al. [25] reported that the cytoskeleton is anchored to the cell membrane, where surface-bound condensates exert and buffer mechanical forces, thereby conferring both stability and flexibility to the cytoskeletal structure. During skeletal muscle development, *TPM1* promotes sarcomere assembly by stabilizing actin filaments and regulates the differentiation of myocytes into functional fast-twitch (Type II) and slow-twitch (Type I) muscle fibers [26]. Human studies indicate that the expression levels of *TPM1* directly influence the calcium sensitivity and contraction characteristics of muscle fibers [27]. This study found that in the longissimus dorsi muscle group, the expression of *TPM1* was significantly higher in stallions than in mares, with marked enrichment in the cytoskeleton in muscle cells pathway. Muscle phenotype data further indicated that the average muscle fiber area in stallions exceeded that in mares. It has been reported that male animals generally exhibit larger average muscle fiber areas than females, which is consistent with the findings of this study. Sex hormones also play a pivotal role in maintaining skeletal muscle homeostasis. Among them, testosterone acts as a potent anabolic factor promoting protein synthesis and muscle regeneration [28]. Therefore, it is hypothesized that muscle fiber thickness in horses may be sex-dependent. *MYL1*, a member of the myosin light chain family, is essential for maintaining skeletal muscle function [29]. Feng et al. [30] discovered that aerobic and resistance exercise alleviate skeletal muscle atrophy in myocardial infarction (MI) models through the IGF-1/IGF-1R-PI3K/Akt signaling pathway by inhibiting apoptosis. Horses possess exceptional aerobic and muscular abilities, excelling in competitive sports. Murach KA et al. [31] proposed a synergistic role for *TPM1* and *MYL1*: *TPM1* exposes actin binding sites via calcium signaling, and *MYL1* enhances the myosin head’s motility, ensuring coordinated muscle contraction. *MYH3* is a member of the myosin family, which consists of two myosin heavy chains and four light chains [32]. It functions as a motor protein in muscle contraction, cell migration, and cytokinesis. By hydrolyzing ATP, *MYH3* converts chemical energy into mechanical energy and contributes to muscle contraction, making it a critical component of muscle tissue [33]. *MYH3* impacts muscle development and regulates downstream protein phosphorylation via MAPK and TGF-β signaling pathways, thereby influencing muscle metabolism [34]. In this study, diaphragm muscle expression of *MYL1* and *MYH3* was significantly higher in mares than in stallions, with both genes markedly enriched in the cytoskeleton in muscle cells pathway. Amino acid composition analysis revealed that glycine content in the diaphragm of mares significantly exceeded that of stallions. As a precursor of glutathione (GSH), glycine supplementation can indirectly enhance antioxidant capacity and reduce oxidative damage [35]. Chemello et al. [36] reported higher *MYH3* expression in the soleus muscle than in the extensor digitorum longus muscle. Estrogen has been proven to modulate muscle mass, function, and antioxidant capacity [37]. Therefore, it is hypothesized that sex may influence the antioxidant capacity of equine muscle.

PPI network construction identified multiple DEGs, including *TPM1*, *MYL1*, *MYH3* and *PYGM*. *PYGM* catalyzes the degradation of glycogen to glucose-1-phosphate, providing energy by breaking down glycogen stored in muscle tissue [38]. This study found that in the longissimus dorsi muscle group, the expression of *PYGM* was significantly higher in stallions than in mares, suggesting superior energy-supplying capacity in stallions. Nam et al. [39] examined the genomic structure and expression patterns of *PYGM* in thoroughbred horses and found that *PYGM* expression was highest in skeletal muscle compared to other tissues. Sun [40] identified DEGs, including *TPM1*, *MYL1*, *MYH3* and *PYGM*, co-expressed in the longissimus dorsi muscle of donkeys. These genes influence skeletal muscle development by modifying muscle fiber contraction and protein metabolism. This study demonstrated that the sex-specific differential expression of *TPM1*, *MYL1*, *MYH3* and *PYGM* significantly impacts muscle structure, thereby providing a molecular-level explanation for observed sex-related differences. The differential expression of *TPM1*, *MYL1* and *MYH3* suggests that the fiber type composition (i.e., the proportion of type I and type II fibers) may vary with Kazakh horse genders. However, given insufficient available data on fiber types in Kazakh horses, this study did not directly assess fiber type distribution. These transcriptomic findings imply potential sex-related differences in muscle fiber types, which warrant further validation through approaches such as immunohistochemistry of specific fiber types and single-fiber RNA sequencing.

In this study, the use of paraffin embedding and sectioning may introduce a certain degree of tissue shrinkage, potentially affecting the clarity of morphological observations. Although incomplete fibers were excluded during image analysis to ensure the reliability of quantitative data, future studies employing cryosectioning or alternative techniques that better preserve the native tissue architecture will provide further verification of these findings. Despite the lack of systematic multi-omics analyses, consistent patterns were observed between metabolic phenotypes and gene expression. For instance, sex-specific differences in fatty acid composition may reflect differential regulation of lipid metabolism genes. The elevated C18:2n6c content in stallion rectus abdominis may indicate an increased demand for glycine synthesis in the diaphragm. These variations may be associated with the expression of genes involved in muscle fiber remodeling. Future studies will integrate multiple data types to comprehensively analyze muscle fiber typing and multi-omics.

## 5. Conclusions

This study conducted transcriptomic profiling of the longissimus dorsi, rectus abdominis, and diaphragm muscles of stallion and mare Kazakh horses. The results demonstrated that DEGs such as *TPM1*, *MYL1*, *MYH3* and *PYGM* are actively involved in pathways such as the cytoskeleton in muscle cells and thyroid hormone, underscoring that these genes may serve as candidate regulators of skeletal muscle growth and development in Kazakh horses.

## Figures and Tables

**Figure 1 biology-14-01216-f001:**
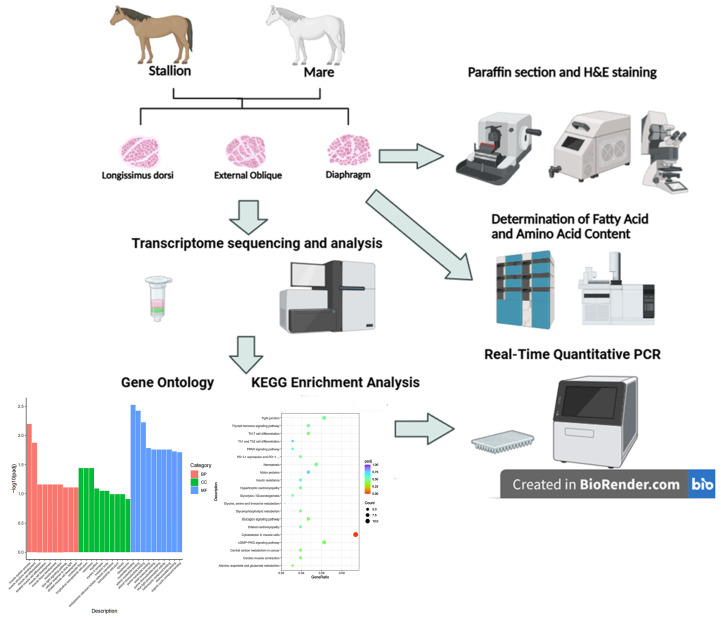
Technology roadmap. **Note:** this figure was created using https://app.biorender.com (accessed on 15 August 2025).

**Figure 2 biology-14-01216-f002:**
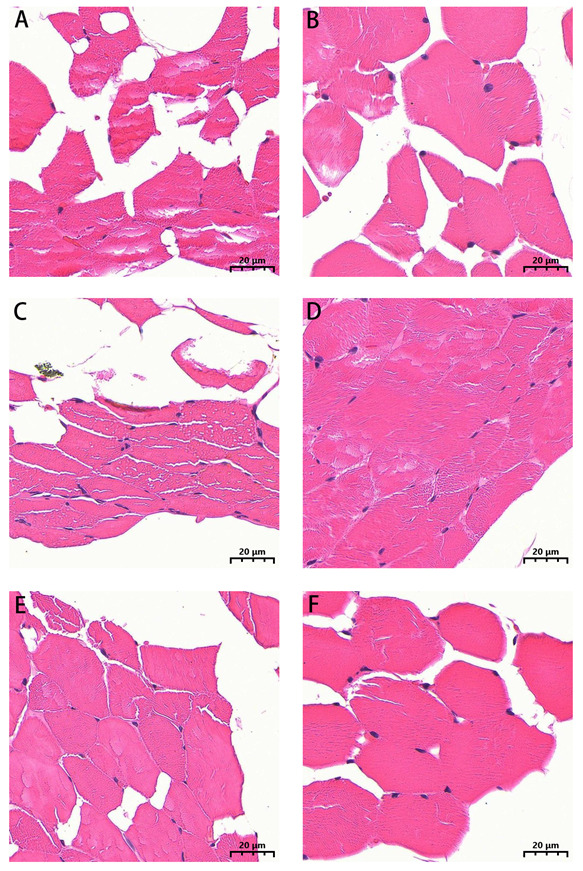
HE staining of the longissimus dorsi, rectus abdominis, and the diaphragm of stallion and mare Kazakh horses (50×). (**A**) The longissimus dorsi of stallion Kazakh horses; (**B**) the longissimus dorsi of mare Kazakh horses; (**C**) the rectus abdominis of stallion Kazakh horses; (**D**) the rectus abdominis of mare Kazakh horses; (**E**) the diaphragm of stallion Kazakh horses; (**F**) the diaphragm of mare Kazakh horses.

**Figure 3 biology-14-01216-f003:**
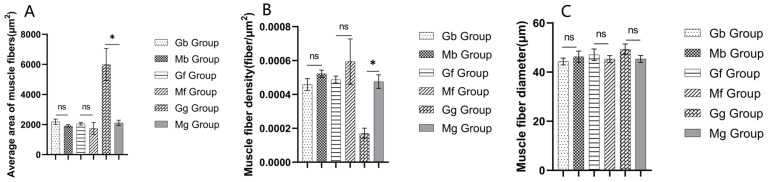
Differences in the average area of muscle fibers, muscle fiber density, and muscle fiber diameter in Gb, Mb, Gf, Mf, Gg, and Mg groups. (**A**) Differences in the average area of fibers; (**B**) differences in muscle fiber density; (**C**) differences in muscle fiber diameter. **Note:** The asterisk (*) indicate significant differences (*p* < 0.05), while ns indicate no significant difference (*p* > 0.05).

**Figure 4 biology-14-01216-f004:**
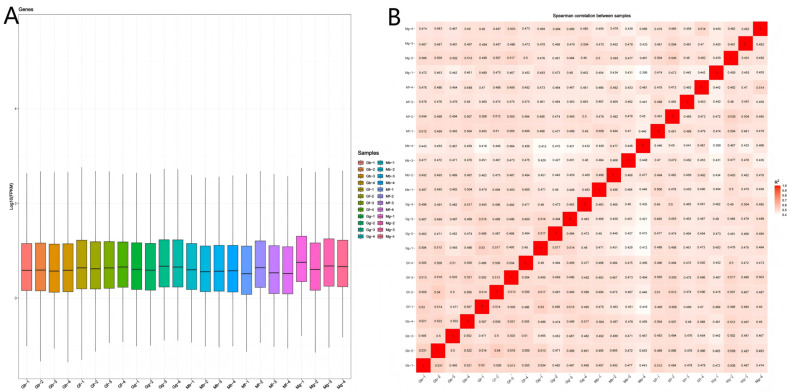
Boxplot of expression levels and correlation heatmap for the Mb, Mf, Mg, Gb, Gf, and Gg groups. (**A**) Boxplot of expression levels across the samples; (**B**) correlation heatmap for the groups.

**Figure 5 biology-14-01216-f005:**
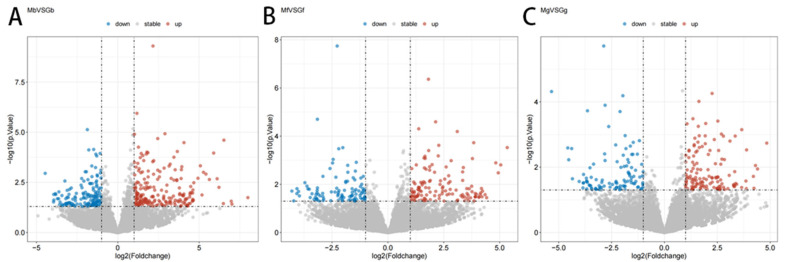
Volcano plots of DEGs. (**A**) Volcano map of the Mb and Gb groups; (**B**) volcano map of the Mf and Gf groups; (**C**) volcano map of the Mg and Gg groups. Note: in the figures, “up” and “down” represent upregulated and downregulated genes, respectively.

**Figure 6 biology-14-01216-f006:**
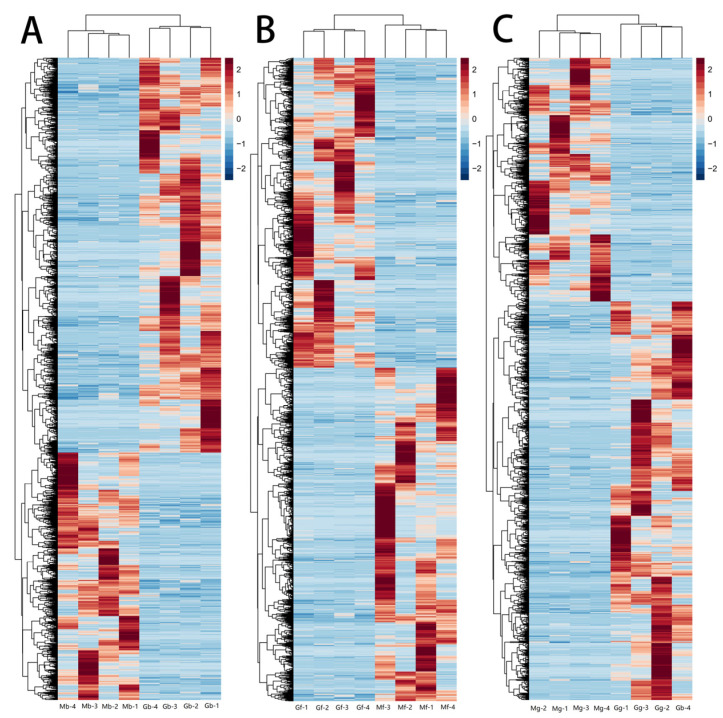
Clustering analysis of DEGs. (**A**) Clustering analysis chart of the Mb and Gb groups; (**B**) clustering analysis chart of the Mf and Gf groups; (**C**) clustering analysis chart of the Mg and Gg groups. Note: the *x*-axis denotes individual samples, and the *y*-axis represents expression levels. The color gradient from blue to red indicates increasing upregulation.

**Figure 7 biology-14-01216-f007:**
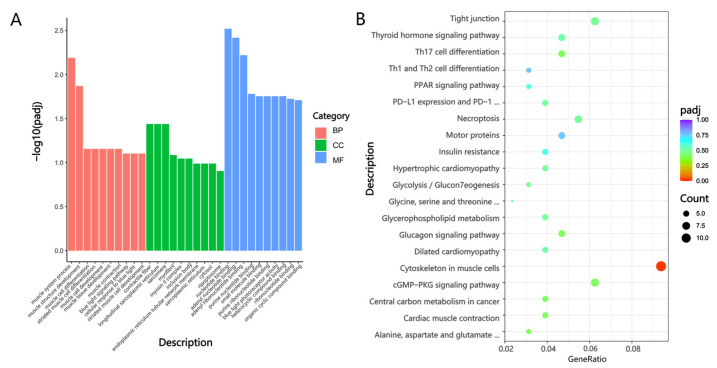
GO annotation and KEGG enrichment analysis in the Mb and Gb Groups. (**A**) GO annotation of DEGs in the Mb and Gb groups; (**B**) GO annotation of DEGs in the Mf and Gf groups.

**Figure 8 biology-14-01216-f008:**
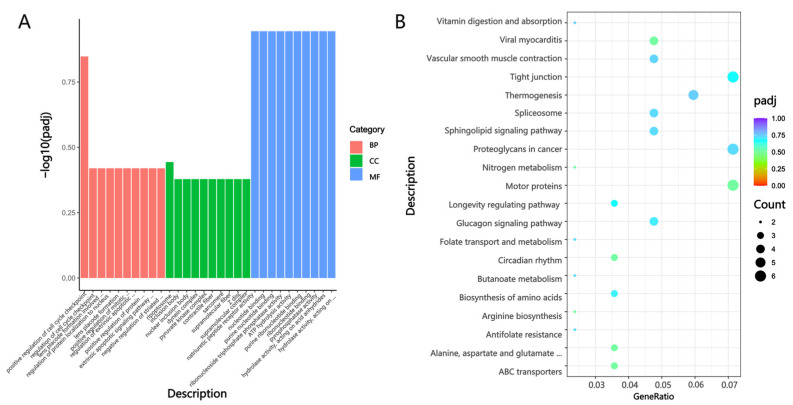
GO annotation and KEGG enrichment analysis in the Mf and Gf Groups. (**A**) GO annotation of DEGs in the Mf and Gf groups; (**B**) KEGG pathway enrichment analysis of DEGs in the Mf and Gf groups.

**Figure 9 biology-14-01216-f009:**
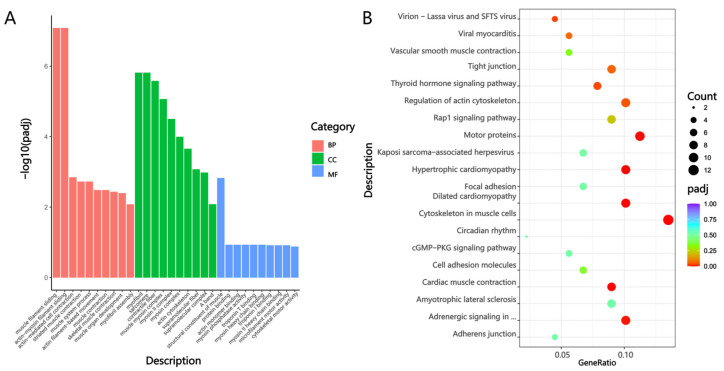
GO annotation and KEGG enrichment analysis in the Mg and Gg Groups. (**A**) GO annotation of DEGs in the Mg and Gg groups; (**B**) KEGG pathway enrichment analysis of DEGs in the Mg and Gg groups. Note: in Figure 7A, Figure 8A and Figure 9A, the *x*-axis represents DEGs, with red indicating biological process (BP), green denoting cellular component (CC), and blue signifying molecular function (MF). Figure 7B, Figure 8B and Figure 9B illustrate the top 20 pathways with the lowest Q-values. The *y*-axis shows the pathway names, and the *x*-axis represents the gene ratio. The “count” denotes the quantity, and the color gradient from blue to red indicates decreasing Q-values.

**Figure 10 biology-14-01216-f010:**
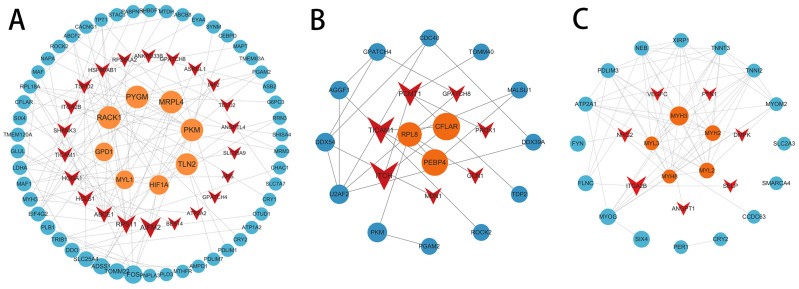
PPI network of DEGs and identified core genes. (**A**) PPI of Mb vs. Gb; (**B**) PPI of Mf vs. Gf; (**C**) PPI of Mg vs. Gg. Circles represent upregulated genes, and triangles denote downregulated genes.

**Figure 11 biology-14-01216-f011:**
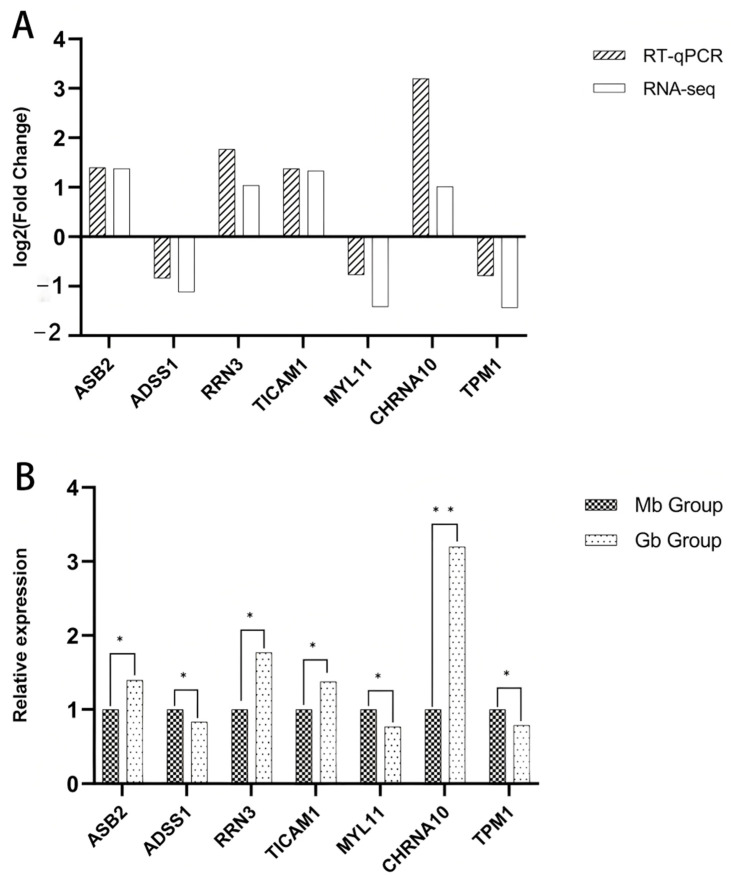
RT-qPCR validation. Validation of differentially expressed genes by RT-qPCR. (**A**) Log_2_ fold-change comparison between RNA-seq and RT-qPCR for differentially expressed genes; (**B**) relative expression of differentially expressed genes by RT-qPCR. **Note:** The asterisk (*) indicate significant differences (*p* < 0.05), while the double asterisk (**) indicate highly significant discrepancies (*p* < 0.01).

**Table 1 biology-14-01216-t001:** Comparison of fatty acid composition in three parts of stallion and mare horses (ng/mg).

Fatty Acid	The Longissimus Dorsi		The Rectus Abdominis		The Diaphragm	
	Stallion	Mare	Stallion	Mare	Stallion	Mare
C6:0	0.05 ± 0.03	0.04 ± 0.02	0.09 ± 0.09	0.05 ± 0.03	0.04 ± 0.03	0.06 ± 0.03
C8:0	0.12 ± 0.08	0.12 ± 0.08	0.22 ± 0.21	0.20 ± 0.14	0.10 ± 0.07	0.20 ± 0.13
C10:0	0.86 ± 0.69	0.78 ± 0.54	1.38 ± 1.13	1.25 ± 0.82	0.651 ± 0.410	1.304 ± 0.838
C11:0	0.02 ± 0.01	0.02 ± 0.01	0.04 ± 0.04	0.03 ± 0.02	0.03 ± 0.02	0.04 ± 0.02
C18:2n6c	109.43 ± 31.54	86.92 ± 9.28	152.27 ± 83.81 ^A^	134.23 ± 26.20 ^B^	144.27 ± 39.60	146.44 ± 49.47
C18:3n3	28.40 ± 18.19 ^A^	16.05 ± 3.03 ^B^	66.27 ± 48.38	42.37 ± 39.26	40.20 ± 17.54	30.87 ± 18.13
C20:2	1.97 ± 0.94	1.57 ± 0.26	3.58 ± 2.23 ^A^	3.04 ± 0.77 ^B^	2.99 ± 1.11	3.39 ± 1.58
C20:3n3	1.51 ± 0.75 ^A^	0.83 ± 0.22 ^B^	3.51 ± 1.68	2.12 ± 1.44	2.65 ± 0.90	2.09 ± 1.29
C24:0	0.25 ± 0.06	0.22 ± 0.03	0.40 ± 0.17 ^a^	0.30 ± 0.07 ^b^	0.36 ± 0.08	0.40 ± 0.09

Note: Data are expressed as mean ± SD (*n* = 4). Letters a and b indicate significant differences (*p* < 0.05), while letters A and B indicate highly significant discrepancies (*p* < 0.01).

**Table 2 biology-14-01216-t002:** Comparison of amino acid composition in three parts of stallion and mare horses (ug/mg).

Amino Acid	The Longissimus Dorsi		The Rectus Abdominis		The Diaphragm	
	Stallion	Mare	Stallion	Mare	Stallion	Mare
Gly	14.22 ± 6.49	10.79 ± 2.62	14.22 ± 6.49	10.79 ± 2.62	7.42 ± 1.69 ^B^	9.56 ± 3.46 ^A^
Ala	15.41 ± 6.045 ^A^	13.39 ± 1.30 ^B^	15.41 ± 6.04 ^A^	13.39 ± 1.30 ^B^	9.01 ± 2.09	11.25 ± 2.03
Ser	25.47 ± 13.98	16.98 ± 9.08	25.47 ± 13.98	16.98 ± 9.08	5.39 ± 1.71	8.80 ± 4.09
Pro	8.20 ± 2.23 ^A^	7.51 ± 0.54 ^B^	8.20 ± 2.23 ^A^	7.51 ± 0.54 ^B^	7.49 ± 1.48	7.77 ± 1.64
Val	7.07 ± 4.91	7.01 ± 1.81	7.07 ± 4.91	7.01 ± 1.81	2.78 ± 2.16	3.94 ± 1.60
Thr	9.97 ± 2.65	9.13 ± 1.20	9.97 ± 2.65	9.13 ± 1.20	6.63 ± 0.99	7.88 ± 0.94
Cys	2.52 ± 0.88	1.65 ± 0.51	2.52 ± 0.88	1.65 ± 0.51	1.94 ± 0.43	2.33 ± 0.59
Ile	9.42 ± 1.72	9.18 ± 1.19	9.42 ± 1.72	9.18 ± 1.19	7.27 ± 1.36	7.64 ± 0.83
Asp	19.65 ± 4.48	24.19 ± 6.25	19.65 ± 4.48	24.19 ± 6.25	13.20 ± 1.07	19.02 ± 3.65
Glu	18.24 ± 1.61	20.59 ± 2.44	18.24 ± 1.61	20.59 ± 2.44	17.46 ± 2.43	20.13 ± 2.26

Note: Data are expressed as mean ± SD (*n* = 4), while A and B denote highly significant discrepancies (*p* < 0.01).

**Table 3 biology-14-01216-t003:** Overall detection of mRNA sequencing data.

Samples	Raw Data	Clean Data	Q20/(%)	Q30/(%)	GC Content/(%)	Mapped Reads
Gb-1	52600700	51228464 (97.39%)	98.59	95.69	51.90	46831680 (91.42%)
Gb-2	60924228	59514598 (97.69%)	98.64	95.84	51.63	54126030 (90.95%)
Gb-3	62928188	61088904 (97.08%)	98.65	95.87	52.63	55202427 (90.36%)
Gb-4	61793134	60379646 (97.71%)	98.64	95.85	51.82	55184736 (91.40%)
Gf-1	61411500	59943692 (97.61%)	98.61	95.73	51.21	54954663 (91.68%)
Gf-2	55196862	53930098 (97.71%)	98.64	95.85	51.32	49656683 (92.08%)
Gf-3	51898192	50491302 (97.29%)	98.50	95.42	52.18	46195189 (91.49%)
Gf-4	44219856	43104312 (97.48%)	98.64	95.83	52.18	39584418 (91.83%)
Gg-1	60469756	58982602 (97.54%)	98.62	95.78	51.33	54006313 (91.56%)
Gg-2	54807072	53441718 (97.51%)	98.57	95.61	51.97	49034062 (91.75%)
Gg-3	53987080	52746006 (97.70%)	98.65	95.86	51.40	48074178 (91.14%)
Gg-4	54524910	53203350 (97.58%)	98.63	95.80	51.23	48404886 (90.98%)
Mb-1	44171652	42999262 (97.35%)	98.26	94.60	52.05	39358453 (91.53%)
Mb-2	49446994	48218616 (97.52%)	98.59	95.67	52.48	43992403 (91.24%)
Mb-3	43674054	42368106 (97.01%)	98.26	94.61	52.74	38635399 (91.19%)
Mb-4	45177698	43972694 (97.33%)	98.30	94.74	52.34	40137588 (91.28%)
Mf-1	67470674	65629376 (97.27%)	98.61	95.76	52.47	59982138 (91.40%)
Mf-2	58746810	57378112 (97.67%)	98.64	95.82	51.72	52322210 (91.19%)
Mf-3	59379442	57862616 (97.45%)	98.60	95.71	53.08	53083623 (91.74%)
Mf-4	60953306	59404158 (97.46%)	98.55	95.58	52.43	54483654 (91.72%)
Mg-1	38965048	38011508 (97.55%)	98.31	94.75	50.82	34627582 (91.10%)
Mg-2	52499628	51220078 (97.56%)	98.63	95.81	52.65	46821285 (91.41%)
Mg-3	50485948	49309776 (97.67%)	98.63	95.81	51.16	45026638 (91.31%)
Mg-4	52882244	51478352 (97.35%)	98.55	95.53	51.57	47005951 (91.31%)

Note: Gb group (Gb-1 to Gb-4), Gf group (Gf-1 to Gf-4), Gg group (Gg-1 to Gg-4), Mb group (Mb-1 to Mb-4), Mf group (Mf-1 to Mf-4), Mg group (Mg-1 to Mg-4). Raw data: original sequencing data; clean data: filter the data; Q20: proportion of bases with quality value greater than or equal to 20; Q30: proportion of bases with quality value greater than or equal to 30; GC content: calculate the percentage of the total number of bases G and C to the total number of bases; mapped reads: comparing reads to the genome.

**Table 4 biology-14-01216-t004:** Top 10 DEGs between Mb and Gb groups.

Gene Symbol	Gene Description	*p*-Value	Log2Fold Change
*CHKB*	Choline kinase beta.	7.4200 × 10^−6^	−1.8738
*WIPI1*	phosphoinositide interacting 1	1.18754 × 10^−5^	2.8973
*DMPK*	DM1 protein kinase.	7.24243 × 10^−5^	−1.4851
*ATP2A2*	Calcium-transporting ATPase	0.000114276	−1.0901
*DNAH12*	Uncharacterized protein	0.000123315	1.7507
*NAA38*	N(alpha)-acetyltransferase 38	0.00012399	−1.3493
*TPM1*	Tropomyosin 1	0.000114256	−1.4378
*ZNF704*	Zinc finger protein 704	0.000105586	1.8497
*MYL1*	Myosin light chain 1	0.000145316	1.5398
*ASB2*	Ankyrin repeat and SOCS box containing 2	0.00018393	1.3821

**Table 5 biology-14-01216-t005:** Top 10 DEGs between Mf and Gf groups.

Gene Symbol	Gene Description	*p*-Value	Log2Fold Change
*HSPB2*	SHSP domain-containing protein	1.80506 × 10^−8^	−2.2699
*CSNK1D*	Casein kinase 1 delta	4.33327 × 10^−7^	1.8137
*PROX1*	Prospero homeobox 1	1.98606 × 10^−5^	−3.1549
*GSS*	Glutathione synthetase	4.94602 × 10^−5^	1.3778
*RCE1*	Ras-converting CAAX endopeptidase 1	0.000240301	2.2765
*TPR*	Nuclear basket protein	0.00029971	−2.0182
*CLDND2*	Claudin domain containing 2	0.000333195	−2.1989
*RUFY3*	RUN and FYVE domain containing 3	0.000396518	1.8234
*RNF150*	Ring finger protein 150	0.000657863	2.2527
*PGAM2*	Phosphoglycerate mutase	0.000838227	1.6703

**Table 6 biology-14-01216-t006:** Top 10 DEGs between Mg and Gg groups.

Gene Symbol	Gene Description	*p*-Value	Log2Fold Change
*OSCP1*	Organic solute carrier partner 1	1.95223 × 10^−6^	−2.8698
*UCKL1*	Zinc finger protein 512B	9.68441 × 10^−5^	1.6301
*MTMR7*	Myotubularin-related protein 7	0.000127264	−2.8051
*ANGPTL4*	Angiopoietin-like 4	0.000198384	−2.0992
*MYH2*	Myosin-2; muscle contraction	0.000329082	1.3481
*MAF*	MAF bZIP transcription factor	0.000460435	1.9523
*TPM1*	Tropomyosin 1	0.000562281	1.6019
*MYH3*	Myosin heavy chain 3	0.001120483	2.788
*IGF2BP3*	Insulin-like growth factor 2 mRNA binding protein 3	0.001297589	1.439
*CLDN6*	Through calcium-independent cell-adhesion activity	0.001712027	1.5833

## Data Availability

The data presented in this study are openly available in BioProject with reference number PRJNA1261704.

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
