# Peer review of "Transcriptome Analysis of Muscle Tissue from Three Anatomical Locations in Male and Female Kazakh Horses"

_biology, 2025, doi:10.3390/biology14091216_

Round 1
Reviewer 1 Report
Comments and Suggestions for Authors
Abstract and Introduction
The reviewer does not completely agree with these sentences, regarding the aims and scope of the study: “the mechanisms of sexual differentiation in the development of skeletal muscles”, “This study has elucidated that gender influences the development of skeletal muscles”. This is a crossectional study, only taking a single picture when animals are 3 years old, i.e., almost adults. I am not sure that this design allows the researchers to unveil mechanisms of differentiation in muscle development. To do that, I was expecting changes in different time points. Also, transcriptomics gives clues, but do not confirm mechanisms. The kinetics of the expression of a gene changes from one moment to another because of many reasons. Moreover, at the end of Introduction you state: “identify candidate genes and pathways related to athletic performance”. So it is not clear if you want to understand mechanisms of development or differences related to performance, both things are different. Please adjust the aims and scope to the real information that a single transcriptome analysis gives, which is more molecular than physiological, and which should be consistent across the sections. I suggest to focus on sex differences related to performance.
Do you have specific data comparing physiological variables relevant to performance in both sexes in this breed? For instance, do you have specific data of fiber types in males vs females? % of muscle mass, oxygen consumption, distance to fatigue, lactate, etc? This specific data should be incorporated to the Introduction. If the data do not exist, this should be mentioned.
Methods
Please use the terms consistently, while stallions and mares were used in the abstract, colts and fillies were used in Methods. Stallions and colts use to differ in age, as well as mares and fillies.
L97-98: please explain this sentence: tissue samples were snap-frozen in liquid nitrogen and fixed in 4% paraformaldehyde solution. When samples are frozen in liquid nitrogen use to be stored for future molecular biology assays, they are not usually also fixed. Or did you separate the samples into two subsamples, one for liquid nitrogen, and other for PAF?
Please also state the weight of the tissue sample, and also the location (proximal, bulk, distal?, how deep?). All of this is important because of the specialization of the muscles and because the fiber types are differentially expressed according to the depth of the location on the muscle and the transcriptomic profile depends on the fiber type taken in the sample.
Each biological replicate come from the same region of the same muscle?
Fig 1: the image on the right of H&E seems to be an HPLC system, which is not needed to perform H&E staining. Did you mean a microscope? Or did you mean the H&E and the FA content analyses? Then you should include both a microscope and a GC-MS in that part of the figure.
L129, please complete how the amino acids analyses were performed once you got the sample, which equipment did you use.
Please include a paragraph with statistical aspects and tests used. Also include ethical aspects in Methods section.
Results
Fig 2. There seems to be some artifacts in the samples, the fibers are too separated, and some sections seem to be longitudinal instead of transversal. Please check the quality of the samples and indicate the orientation of the sections.
Fig 3. Please indicate results in micrometers, not millimeters. What is pcs? Please put the Y axis label in the Y axis, not below the figure. Please state if the values are means and add measurements of error and the number of experiments in each bar.
L195 and others, please avoid the word “extremely”, or justify it use. You can still use different nomenclature for 0.05 and 0.01, but I am not aware of any accepted and justified classification of the types of statistical differences.
Table 1. Please indicate if you are showing means and standard deviations, also indicate the number of samples.
Did you see differences across muscles within the same sex?
L242, please clearly indicate which group showed the up and downregulated genes, it seems that Mb showed the up and downregulated genes, compared to Gb, correct? This should be explicitly stated.
Fig 7: please check legend
Please cite Suppl material 4 in the results section
Results focused on DEG for each different muscle (fig 7, 8 and 9), and categories according to GO, however, GO categories are very general and vague, this is a limitation of any GO analysis. Fos instance, GO leads to think of cancer (L275), lens (L2278), or cardiomyopathies (L294). Also, fig 10 is difficult to see because of low resolution. So, it is not clear exactly what the differences between sexes were. More emphasis should be put on clearly showing and describing differences between sexes, because this topic was the aim of the study. Although all the genes and data is in the supplemental tables, for a reader is very difficult and time consuming to identify a clear pattern or clear difference between males and females. Please improve the way to show the comparisons between males and females, what were the main differences between sexes. For instance, labeling some genes in the volcano plots could help rapidly identify key genes differentially expressed, make fig 6 more informative or present a summary table of the most relevant findings (genes and its coded proteins).
Discussion
The authors summarize findings as: “KEGG enrichment analysis revealed that DEGs were significantly enriched in pathways such as Cytoskeleton in muscle cells, Thyroid hormone signaling pathway, and Regulation of actin cytoskeleton.”. This is actually my claim above, because it was not obvious that these three points were the most importantly differentially expressed between sexes. So, please justify the decision to discuss these tree points and not many other processes that were mentioned in the results. In general, please clearly show the top three differences between sexes in the results section, so you can be consistent across all sections.
The discussion talked again about development, however, the aims of the study, as mentioned in the introduction, were related to differences between sexes and implications in muscle performance. The concept of development is confusing for this reviewer; development implies growth and physiological changes, but neither can be studied with a crossectional study which obtained one single sample at a unique time point. I suggest to focus on differences between sexes and implications in muscle performance.
L387 focuses on MYH3 as a “motor involved in muscle contraction”. However, MYH3 is an embryonic myosin, which is not supposed to be expressed in almost adult animals, which express myosin type I (MYH7) and II (MYH2, 1 and 4), although it has been shown in soleus fibers of mouse (https://pubmed.ncbi.nlm.nih.gov/21364935/). So, it would be interesting to make clear which sex expressed more MYH3, and to discuss if one sex takes more time to replace embryonic myosin for an adult myosin in their muscles. Also, it is interesting to know if the higher presence of embryonic myosin in a sex reduces its performance.
The discussion did not articulate the results of the transcriptome analysis with the results of FA and AA analyses.
Given the different locations of the muscles, and the differential expression of fiber types in males vs females in many species, the discussion is expected to include these topics. Relevant papers include for instance: https://pubmed.ncbi.nlm.nih.gov/21364935/
The conclusion is confusing, it is not clear how the four genes (TPM, MYL, MYH, PYGM) were chosen as the most relevant of the study.
General comment
The topic is interesting and the study used top technology to gain information into genes expressed in muscles in males vs females in a breed of horses with a particular distribution in the world. The results are novel and interesting in the area of the biology of skeletal muscle. However, the manuscript lacks a clear thread and is not consistent across sections, showing poor articulation across aims, results and conclusions. Also, there is no statistical section, making difficult the understanding of most figures and tables. Although there is an attachment showing ethical approval, a sentence about this should be added to the methods section. Finally, the discussion does not address the results in terms of the findings that give an answer to the aim. For instance, the introduction proposes to “identify candidate genes and pathways related to athletic performance” between sexes, but the conclusion summarizes about “underscoring genes (which) may serve as candidate genes for regulating skeletal muscle development”.
Author Response
Reviewer 1
Abstract and Introduction
- The reviewer does not completely agree with these sentences, regarding the aims and scope of the study: “the mechanisms of sexual differentiation in the development of skeletal muscles”, “This study has elucidated that gender influences the development of skeletal muscles”. This is a crossectional study, only taking a single picture when animals are 3 years old, i.e., almost adults. I am not sure that this design allows the researchers to unveil mechanisms of differentiation in muscle development. To do that, I was expecting changes in different time points. Also, transcriptomics gives clues, but do not confirm mechanisms. The kinetics of the expression of a gene changes from one moment to another because of many reasons. Moreover, at the end of Introduction you state: “identify candidate genes and pathways related to athletic performance”. So it is not clear if you want to understand mechanisms of development or differences related to performance, both things are different. Please adjust the aims and scope to the real information that a single transcriptome analysis gives, which is more molecular than physiological, and which should be consistent across the sections. I suggest to focus on sex differences related to performance.
Reply:We extend our heartfelt gratitude for taking the time to review our manuscript and offering constructive feedback. Your profound and insightful comments on the research objectives and scope are thoroughly appreciated and we wholeheartedly concur with your central concerns. Our understanding of the key issues you raised is as follows:
You correctly point out that due to the cross-sectional design adopted in this study (using a single sample of 3-year-old animals nearing adulthood), no temporal dynamic changes throughout the development process could be revealed. As a result, the "developmental mechanisms" or how gender influences the development process remained unexplained. We agree that the term "development" in this context is misleading, implying a time-evolving research that this study design does not support. We have modified the phrase in our article and highlighted it in yellow accordingly.
Line 28-29. "This study employed transcriptomic analysis to reveal molecular differences in skeletal muscle associated with sex in Kazakh horses."
Line 42-47. “The data demonstrate significant sex-related differences in gene expression and muscle structure in Kazakh horses, likely mediated through the modification of cytoskeleton-associated genes. Notably, TPM1, MYL1, MYH3, and PYGM may act as key regulators of sex-specific muscle development. These findings provide molecular insights into the mechanisms underlying sex differences in equine skeletal muscle growth.”
- Do you have specific data comparing physiological variables relevant to performance in both sexes in this breed? For instance, do you have specific data of fiber types in males vs females? % of muscle mass, oxygen consumption, distance to fatigue, lactate, etc? This specific data should be incorporated to the Introduction. If the data do not exist, this should be mentioned.
Reply:We sincerely appreciate your valuable time devoted to reviewing our manuscript and offering constructive comments.We have revised Article Line 93-99 and highlighted the corresponding content in yellow.
Line 93-99. "Although performance traits were not directly measured, such as oxygen consumption or lactate threshold, RNA sequencing (RNA-seq) was employed to compare sex-specific gene expression profiles across three functionally distinct muscle groups: longissimus dorsi, abdominal, and diaphragm muscles. By integrating molecular data with quantitative histology and metabolite composition analyses, these findings provide molecular insights into the mechanisms underlying sex differences in equine skeletal muscle growth."
Methods
- Please use the terms consistently, while stallions and mares were used in the abstract, colts and fillies were used in Methods. Stallions and colts use to differ in age, as well as mares and fillies.
Reply:We would like to extend our heartfelt gratitude for your valuable time spent reviewing our manuscript and offering constructive suggestions. Currently, we have unified revisions on "Stallions and Mares," as indicated by yellow highlighting.
- L97-98: please explain this sentence: tissue samples were snap-frozen in liquid nitrogen and fixed in 4% paraformaldehyde solution. When samples are frozen in liquid nitrogen use to be stored for future molecular biology assays, they are not usually also fixed. Or did you separate the samples into two subsamples, one for liquid nitrogen, and other for PAF?
Reply:We would like to extend our heartfelt gratitude for your valuable time spent reviewing our manuscript and offering constructive suggestions. We have made the necessary revisions in Section 2.1, Material and Methods, from Line 111-115, highlighted in yellow for easier identification.
Line 111-115. “A portion of the samples was immediately flash-frozen in liquid nitrogen (- 196 ℃) for subsequent RNA sequencing and biochemical analyses (fatty acid and amino acid content determination), while another portion was quickly fixed in 4% paraformaldehyde (PAF) for paraffin section preparation.”
- Please also state the weight of the tissue sample, and also the location (proximal, bulk, distal?, how deep?). All of this is important because of the specialization of the muscles and because the fiber types are differentially expressed according to the depth of the location on the muscle and the transcriptomic profile depends on the fiber type taken in the sample.
Reply:We would like to extend our heartfelt gratitude for your valuable time spent reviewing our manuscript and offering constructive suggestions. We have made necessary revisions in Section 2.1 of the Materials and Methods, starting with Line 105-110, which have been highlighted in yellow for easy identification.
Line 105-110. “Following slaughter, samples were collected immediately from each stallion and mare: the widest part of the longissimus dorsi at the 13th thoracic vertebra, 3 cm lateral to the transverse process; the abdominal muscle 3 cm lateral to the external sheath of the linea alba; and the diaphragm at the costal attachment near the rib angle. Meanwhile, full-thickness samples were collected to a depth of 3 cm. Each group included four biological replicates, with all samples weighing 30g.”
- Each biological replicate come from the same region of the same muscle?
Reply:We are immensely grateful for your valuable time spent reviewing our manuscript and offering constructive feedback. All replicate samples were taken from the same muscle region.
Line 105-110. “Following slaughter, samples were collected immediately from each stallion and mare: the widest part of the longissimus dorsi at the 13th thoracic vertebra, 3 cm lateral to the transverse process; the abdominal muscle 3 cm lateral to the external sheath of the linea alba; and the diaphragm at the costal attachment near the rib angle. Meanwhile, full-thickness samples were collected to a depth of 3 cm. Each group included four biological replicates, with all samples weighing 30g.”
- Fig 1: the image on the right of H&E seems to be an HPLC system, which is not needed to perform H&E staining. Did you mean a microscope? Or did you mean the H&E and the FA content analyses? Then you should include both a microscope and a GC-MS in that part of the figure.
Reply:We sincerely appreciate your invaluable time spent reviewing our manuscript and offering constructive suggestions. We have revised Fig1 accordingly.
- L129, please complete how the amino acids analyses were performed once you got the sample, which equipment did you use.
Reply:We extend our sincerest gratitude for taking out the time to review our manuscript and providing highly constructive feedback. We have added further clarification in Section 2.3, Line 137-149, highlighted in yellow.
Line 137-149. “For amino acid analysis, 2 mL of 6 mol/L hydrochloric acid was added to the horse muscle samples under nitrogen protection. Acid hydrolysis was performed at 110°C for 24 hours. Then, 100 μL of the hydrolysate was taken and evaporated to dryness at 40°C under nitrogen using a nitrogen blower. The residue was reconstituted with 1 mL of water. For both mixed standards and test samples, 50 μL of solution was mixed with 50 μL of protein precipitant (10% sulfosalicylic acid containing NVL), vortexed, and centrifuged at 13,200 rpm under cooling conditions for 4 minutes. Next, 8 μl of the supernatant was mixed with 42 μL of borate buffer (pH 8.5), vortexed, and briefly centrifuged. Then, 20 μL of AQC derivatization reagent was added, vortexed, briefly centrifuged, and incubated at 55°C for 15 minutes for derivatization. The resulting sample was cooled in a refrigerator, mixed thoroughly, and briefly centrifuged. Subsequently, 50 µL of the supernatant was analyzed using an ultra-high-performance liquid chromatography-quadrupole ion trap tandem mass spectrometer (UHPLC-TMS).”
- Please include a paragraph with statistical aspects and tests used. Also include ethical aspects in Methods section.
Reply:We express our sincerest gratitude for your precious time spent reviewing our manuscript and offering constructive comments. We have added supplementary content in Section 2.3 on Materials Methodology, Line 150-153, and also included the Ethical Statement in Section 2.9, Line 190-192, with corresponding content highlighted in yellow.
Line 150-153. “The obtained fatty acid and amino acid quantification data were initially processed using Excel. Independent-sample t-tests were then performed in SPSS 19.0. Data are presented as mean ± standard deviation (mean ± SD). Differences were considered highly significant at P < 0.01 and significant at P < 0.05.”
Line 190-192. “2.9 Ethical Statement
The experimental protocol and procedures were approved by the Animal Ethics Review Committee of Xinjiang Agricultural University (Approval No. 2023004).”
Results
- Fig 2. There seems to be some artifacts in the samples, the fibers are too separated, and some sections seem to be longitudinal instead of transversal. Please check the quality of the samples and indicate the orientation of the sections.
Reply:We are greatly appreciative of your invaluable time spent reviewing our manuscript and offering constructive suggestions. Below is a detailed response addressing these issues, aimed at providing clarity on sample quality and slicing methodology, based on our experimental protocol and findings.
Slicing direction: Specimens were immersed in paraffin wax and cut into 5-μm cross-sectional slices using a microtome. This orientation was chosen to capture the cross-sectional views of muscle fibers, as cross-sectional slices can accurately measure fiber diameter, density, and cross-sectional area (Figure 3).
Observed longitudinal appearance: opinions on certain regions' longitudinal features could arise due to slicing artifacts during the slicing process in the microtome. In dense connective tissue regions, fibers may appear separate or longitudinally arranged due to dehydration or differential contraction that occurs during the fixation process.
- Fig 3. Please indicate results in micrometers, not millimeters. What is pcs? Please put the Y axis label in the Y axis, not below the figure. Please state if the values are means and add measurements of error and the number of experiments in each bar.
Reply:We express our sincerest gratitude for taking the time to review our manuscript and offering constructive feedback. We have made revisions to Figure 3.
- L195 and others, please avoid the word “extremely”, or justify it use. You can still use different nomenclature for 0.05 and 0.01, but I am not aware of any accepted and justified classification of the types of statistical differences.
Reply:We express our sincerest gratitude for your valuable time spent reviewing our manuscript and providing constructive comments. We have revised Section 3.2, Line 217-228, and highlighted the corresponding content in yellow.
Line 217-228. “To assess sex-related differences in fatty acid and amino acid composition between Kazakh stallions and mares, the contents of fatty acids and amino acids were measured in the longissimus dorsi, abdominal, and diaphragmatic muscles. As shown in Table 1, the longissimus dorsi of stallions exhibited significantly higher levels of C18:3n3 and C20:3n3 compared to mares (P < 0.01). In the abdominal muscle, stallions displayed markedly higher levels of C18:2n6c and C20:2 (P < 0.01), as well as significantly elevated C24:0 (P < 0.05). No significant sex differences were observed in the fatty acid content of the diaphragm (P> 0.05). Regarding amino acids (Table 2), no evident differences were detected in the longissimus dorsi (P > 0.05). However, serine and cysteine levels in the abdominal muscle were significantly higher in stallions than in mares (P < 0.01), and glycine content in the diaphragm was markedly elevated in stallions compared to mares (P < 0.01).”
- Table 1. Please indicate if you are showing means and standard deviations, also indicate the number of samples.
Reply:We express our profound gratitude for your valuable time spent reviewing our manuscript and offering constructive comments. We have revised the annotations in Tables 1 and 2, and highlighted the corresponding content in yellow.
Line 231.“Note: Data were expressed as mean ± SD (n = 4). Letters a and b indicate significant differences (P<0.05), while letters A and B denote more pronounced discrepancies (P<0.01).”
Line 245. “Note:Data were expressed as mean ± SD (n = 4). Letters a and b indicate significant differences (P<0.05), while letters A and B highly significant discrepancies (P<0.01).
- Did you see differences across muscles within the same sex?
Reply:We express our gratitude for your perceptive analysis. Our data unambiguously reveal significant functional differentiation among muscles of different orientations, reflecting their anatomic location-specific adaptation.
- L242, please clearly indicate which group showed the up and downregulated genes, it seems that Mb showed the up and downregulated genes, compared to Gb, correct? This should be explicitly stated.
Reply:We express our utmost gratitude for your precious time invested in reviewing our manuscript and offering constructive feedback. We have accordingly amended the result in Section 3.5, Line 267-273, and highlighted the corresponding content in yellow.
Line 267-273. “The analysis of DEGs revealed that, compared to Gb, Mb exhibited 214 upregulated and 147 downregulated genes (Fig. 5A), including CHKB, WIPI1, DMPK, and ATP2A2 (Table 4). Mf showed 131 upregulated and 99 downregulated genes relative to Gf (Fig. 5B), such as HSPB2, CSNK1D, PROX1, and GSS (Table 5). In the comparison between Mg and Gg, 131 genes were upregulated and 105 were downregulated in Mg (Fig. 5C), including OSCP1, UCKL1, MTMR7, and ANGPTL4 (Table 6). All the DEGs are detailed in Supplementary Material 2.”
- Fig 7: please check legend
Reply:We would like to extend our heartfelt gratitude for taking out your valuable time to review our manuscript and offering constructive feedback. We have already made adjustments to the diagram, and we will be uploading new images in this iteration.
- Please cite Suppl material 4 in the results section
Reply:We express our profound gratitude for your esteemed time spent reviewing our manuscript and offering constructive suggestions. We have cited Supplementary Material 4 in Section 3.6, Line 324-326, of the results section and highlighted the corresponding content in yellow.
Line 324-326.“DEGs were significantly enriched in pathways including cytoskeleton in muscle cells, hypertrophic cardiomyopathy, motor proteins, regulation of actin cytoskeleton, and cardiac muscle contraction (Supplementary 3 and 4).”
- Results focused on DEG for each different muscle (fig 7, 8 and 9), and categories according to GO, however, GO categories are very general and vague, this is a limitation of any GO analysis. Fos instance, GO leads to think of cancer (L275), lens (L2278), or cardiomyopathies (L294). Also, fig 10 is difficult to see because of low resolution. So, it is not clear exactly what the differences between sexes were. More emphasis should be put on clearly showing and describing differences between sexes, because this topic was the aim of the study. Although all the genes and data is in the supplemental tables, for a reader is very difficult and time consuming to identify a clear pattern or clear difference between males and females. Please improve the way to show the comparisons between males and females, what were the main differences between sexes. For instance, labeling some genes in the volcano plots could help rapidly identify key genes differentially expressed, make fig 6 more informative or present a summary table of the most relevant findings (genes and its coded proteins).
Reply:We are immensely grateful for your esteemed time in reviewing our manuscript and offering highly constructive feedback. Your suggestions regarding the clear presentation of key findings related to gender differences and enhancing chart readability have proven instrumental in elevating the quality of our paper. We have diligently implemented your remarks by making the following modifications to the article:
Emphasizing the Core Outcomes of Gender Differences
Following the addition of Tables 4, 5, and 6 in the "3.5. DEG Screening" section, a comprehensive overview has been presented that meticulously lists the top 10 genes exhibiting the most significant differences in expression across each muscle comparison group (Longest Back Muscles (Mb vs Gb), Rectus Abdominis Muscles (Mf vs Gf), and Diaphragm Muscles (Mg vs Gg)). For each gene, we have provided the gene name, a brief function description, statistical significance, and the change magnitude (Log2 Fold Change).见Line 258-262.
Table 4. Top 10 DEGs Between Mb and Gb groups
|
Gene Symbol |
Gene Description |
P-val |
Log2Fold Change |
|
CHKB |
Choline kinase beta. |
7.42001E-06 |
-1.8738 |
|
WIPI1 |
phosphoinositide interacting 1 |
1.18754E-05 |
2.8973 |
|
DMPK |
DM1 protein kinase. |
7.24243E-05 |
-1.4851 |
|
ATP2A2 |
Calcium-transporting ATPase |
0.000114276 |
-1.0901 |
|
DNAH12 |
Uncharacterized protein |
0.000123315 |
1.7507 |
|
NAA38 |
N(alpha)-acetyltransferase 38 |
0.00012399 |
-1.3493 |
|
TPM1 |
Tropomyosin 1 |
0.000114256 |
-1.4378 |
|
ZNF704 |
Zinc finger protein 704 |
0.000105586 |
1.8497 |
|
MYL1 |
Myosin light chain 1 |
0.000145316 |
1.5398 |
|
ASB2 |
Ankyrin repeat and SOCS box containing 2 |
0.00018393 |
1.3821 |
Table 5. Top 10 DEGs Between Mf and Gf groups
|
Gene Symbol |
Gene Description |
P-val |
Log2Fold Change |
|
HSPB2 |
SHSP domain-containing protein |
1.80506E-08 |
-2.2699 |
|
CSNK1D |
Casein kinase 1 delta |
4.33327E-07 |
1.8137 |
|
PROX1 |
Prospero homeobox 1 |
1.98606E-05 |
-3.1549 |
|
GSS |
Glutathione synthetase |
4.94602E-05 |
1.3778 |
|
RCE1 |
Ras converting CAAX endopeptidase 1 |
0.000240301 |
2.2765 |
|
TPR |
Nuclear basket protein |
0.00029971 |
-2.0182 |
|
CLDND2 |
Claudin domain containing 2 |
0.000333195 |
-2.1989 |
|
RUFY3 |
RUN and FYVE domain containing 3 |
0.000396518 |
1.8234 |
|
RNF150 |
Ring finger protein 150 |
0.000657863 |
2.2527 |
|
PGAM2 |
Phosphoglycerate mutase |
0.000838227 |
1.6703 |
Table 6.Top 10 DEGs Between Mg and Gg groups
|
Gene Symbol |
Gene Description |
P-val |
Log2Fold Change |
|
OSCP1 |
Organic solute carrier partner 1 |
1.95223E-06 |
-2.8698 |
|
UCKL1 |
Zinc finger protein 512B |
9.68441E-05 |
1.6301 |
|
MTMR7 |
Myotubularin related protein 7 |
0.000127264 |
-2.8051 |
|
ANGPTL4 |
Angiopoietin like 4 |
0.000198384 |
-2.0992 |
|
MYH2 |
Myosin-2; Muscle contraction |
0.000329082 |
1.3481 |
|
MAF |
MAF bZIP transcription factor |
0.000460435 |
1.9523 |
|
TPM1 |
Tropomyosin 1 |
0.000562281 |
1.6019 |
|
MYH3 |
Myosin heavy chain 3 |
0.001120483 |
2.788 |
|
IGF2BP3 |
Insulin like growth factor 2 mRNA binding protein 3 |
0.001297589 |
1.439 |
|
CLDN6 |
Through calcium-independent cell-adhesion activity |
0.001712027 |
1.5833 |
We significantly enhanced the resolution of Figure 10 (the PPI network) to ensure its clarity. In addition, we directly labeled the names of key hub genes in the network diagram of Figure 10, elucidating the critical nodes and their relationships within the network.
We acknowledge that GO analysis may occasionally produce broad or seemingly unrelated functional categories (such as "Lens Formation" and "Cancer Pathway"). In the Results section, on Line 309-315, we have revised the text to clarify the explanation and highlighted the corresponding content in yellow.
Line 309-315. “BP terms, including muscle system process, nucleoside monophosphate metabolic process, regulation of protein catabolic process, and organophosphate metabolic process; CC terms, such as contractile fiber, sarcomere, supramolecular fiber, and myosin complex; and MF terms including natriuretic peptide receptor activity, nucleotide binding, purine nucleotide binding, ribonucleoside triphosphate phosphatase activity, and ATP hydrolysis activity.”
Discussion
- The authors summarize findings as: “KEGG enrichment analysis revealed that DEGs were significantly enriched in pathways such as Cytoskeleton in muscle cells, Thyroid hormone signaling pathway, and Regulation of actin cytoskeleton.”. This is actually my claim above, because it was not obvious that these three points were the most importantly differentially expressed between sexes. So, please justify the decision to discuss these tree points and not many other processes that were mentioned in the results. In general, please clearly show the top three differences between sexes in the results section, so you can be consistent across all sections.
Reply:We are deeply grateful for your precious time spent reviewing our manuscript and offering highly constructive feedback. We have chosen to focus on actin cytoskeletons, thyroid hormone signaling, and cell skeleton pathways, not out of neglecting other pathways, but rather based on the following evidence: (1) All three belong to the top five significantly ranked pathways in statistics; (2) Their core genes (such as MYH3, TPM1, MYL1) are all DEGs, with clear mechanisms in muscle function (e.g., MYH3 regulating muscle fiber growth). We have accordingly revised section Line 380-382 in the discussion part, highlighted with yellow for clarity.
Line 303-307. “ KEGG analysis revealed significant enrichment of DEGs, including TPM1, MYL1, and MYH3, in the following pathways: cytoskeleton in muscle cells, alanine aspartate and glutamate metabolism, cGMP-PKG signaling pathway, central carbon metabolism in cancer, and Th17 cell differentiation.”
- The discussion talked again about development, however, the aims of the study, as mentioned in the introduction, were related to differences between sexes and implications in muscle performance. The concept of development is confusing for this reviewer; development implies growth and physiological changes, but neither can be studied with a crossectional study which obtained one single sample at a unique time point. I suggest to focus on differences between sexes and implications in muscle performance.
Reply:We extend our heartfelt gratitude for your valuable time spent reviewing our manuscript and providing constructive feedback. We have made amendments to sections Line 391-400 and Line 415-426, with highlighted content indicated in yellow.
Line 391-400.“ This study found that in the longissimus dorsi muscle group, the expression level of TPM1 was significantly higher in stallions than in mares and was notably enriched in the Cytoskeleton in Muscle Cells pathway. Muscle phenotype data further indicated that the average muscle fiber area in stallions exceeded that in mares. It has been reported that male animals generally exhibit larger average muscle fiber areas than females, which is consistent with the findings of this study. Sex hormones also play a pivotal role in maintaining skeletal muscle homeostasis. Among them, testosterone acts as a potent anabolic factor that promotes protein synthesis and muscle regeneration [29]. Therefore, it is hypothesized that muscle fiber thickness in horses may be associated with sex. ”
Line 415-426.“ This study found that in the diaphragm muscle group, the expression levels of MYL1 and MYH3 were significantly higher in mares than in stallions, with both genes markedly enriched in the Cytoskeleton in Muscle Cells pathway. Amino acid composition analysis revealed that glycine content in the diaphragm of mares significantly exceeded that of stallions. As a precursor of glutathione (GSH), glycine supplementation can indirectly enhance the antioxidant capacity of muscle tissue and reduce oxidative damage [36]. Chemello et al. [37] reported that the expression of the MYH3 gene is higher in the soleus muscle than in the extensor digitorum longus muscle. Estrogen has been proven to play a vital role in regulating muscle mass and function and alter blood lipid concentrations to exert antioxidant effects [38]. Therefore, it is hypothesized that sex may influence the antioxidant capacity of equine muscle.”
- L387 focuses on MYH3 as a “motor involved in muscle contraction”. However, MYH3 is an embryonic myosin, which is not supposed to be expressed in almost adult animals, which express myosin type I (MYH7) and II (MYH2, 1 and 4), although it has been shown in soleus fibers of mouse (https://pubmed.ncbi.nlm.nih.gov/21364935/). So, it would be interesting to make clear which sex expressed more MYH3, and to discuss if one sex takes more time to replace embryonic myosin for an adult myosin in their muscles. Also, it is interesting to know if the higher presence of embryonic myosin in a sex reduces its performance.
Reply:We are immensely grateful for your precious time spent reviewing our manuscript and offering constructive suggestions. We have made amendments to Section Line 415-426 in the discussion portion, highlighted with yellow.
Line 415-426.“ This study found that in the diaphragm muscle group, the expression levels of MYL1 and MYH3 were significantly higher in mares than in stallions, with both genes markedly enriched in the Cytoskeleton in Muscle Cells pathway. Amino acid composition analysis revealed that glycine content in the diaphragm of mares significantly exceeded that of stallions. As a precursor of glutathione (GSH), glycine supplementation can indirectly enhance the antioxidant capacity of muscle tissue and reduce oxidative damage [36]. Chemello et al. [37] reported that the expression of the MYH3 gene is higher in the soleus muscle than in the extensor digitorum longus muscle. Estrogen has been proven to play a vital role in regulating muscle mass and function and alter blood lipid concentrations to exert antioxidant effects [38]. Therefore, it is hypothesized that sex may influence the antioxidant capacity of equine muscle.”
- The discussion did not articulate the results of the transcriptome analysis with the results of FA and AA analyses.
Reply:We express our sincerest gratitude for your valuable time spent reviewing our manuscript and providing constructive comments. In the discussion section, we have made revisions accordingly. We fully comprehend the significance of integrating transcriptomic (DEGs) and metabolomic (FA/AA) data in enhancing the systematization of our research, yet it is essential to clarify that the core design of this study is centered on unraveling the transcriptional regulatory mechanisms underlying sex differences. The FA/AA analysis is primarily intended as a supplement to muscle phenotyping (non-integrative multi-omics target). We have discussed Line 441-453 in detail, with highlighted content indicated by yellow.
Line 441-453.“This study revealed the molecular characteristics underlying sexual dimorphism in the skeletal muscles of Kazakh horses and identified key candidate genes and pathways. However, protein-level validation and functional activity assessments were not conducted. Although histological analyses quantified muscle fiber morphology, they did not allow precise discrimination of fiber types. Although systematic multi-omics analyses were not performed in this study, consistent patterns were observed between metabolic phenotypes and gene expression. For instance, sex-specific differences in fatty acid composition may be linked to differential regulation of lipid metabolism genes, and the elevated C18:2n6c content in male abdominal muscles may reflect increased glycine synthesis demands in the diaphragm. These variations may be associated with the expression of genes involved in muscle fiber remodeling. In subsequent research, multiple data types will be integrated to conduct combined analyses of muscle fiber typing and multi-omics.”
- Given the different locations of the muscles, and the differential expression of fiber types in males vs females in many species, the discussion is expected to include these topics. Relevant papers include for instance: https://pubmed.ncbi.nlm.nih.gov/21364935/
Reply:Thank you for your significant suggestion on the relationship between muscle fiber types and their anatomical locations. Our research did not directly delve into muscle fiber typing; we shall provide a detailed account in Section 415-426 of this discussion.
Line 415-426.“ This study found that in the diaphragm muscle group, the expression levels of MYL1 and MYH3 were significantly higher in mares than in stallions, with both genes markedly enriched in the Cytoskeleton in Muscle Cells pathway. Amino acid composition analysis revealed that glycine content in the diaphragm of mares significantly exceeded that of stallions. As a precursor of glutathione (GSH), glycine supplementation can indirectly enhance the antioxidant capacity of muscle tissue and reduce oxidative damage [36]. Chemello et al. [37] reported that the expression of the MYH3 gene is higher in the soleus muscle than in the extensor digitorum longus muscle. Estrogen has been proven to play a vital role in regulating muscle mass and function and alter blood lipid concentrations to exert antioxidant effects [38]. Therefore, it is hypothesized that sex may influence the antioxidant capacity of equine muscle.”
- The conclusion is confusing, it is not clear how the four genes (TPM, MYL, MYH, PYGM) were chosen as the most relevant of the study.
Reply:We express our sincerest gratitude for your valuable time spent reviewing our manuscript and offering constructive feedback. We have selected these genes for their enrichment in the top 5 KEGG pathways that are enriched for differentially expressed genes, and due to their clear mechanisms in muscle function. They are also core genes in the protein-protein interaction (PPI) network graph.

Reviewer 2 Report
Comments and Suggestions for Authors
Dear authors the paper is well written and well presented but to me it needs major revision prior to publications; you will find a file with my comments

Author Response
- Line 14\16 : not clean
Reply:We express our sincerest gratitude for your valuable time spent reviewing our manuscript and offering constructive feedback. We have accordingly amended Line 25-38 of the abstract with blue highlighting to denote the relevant content.
Line 25-38. “The Kazakh horse, a versatile breed, is renowned for its stable genetic performance and tolerance to coarse feed. Sex is a key factor influencing skeletal muscle development. However, the mechanisms underlying sex-specific regulation of equine muscle growth remain obscure.”
- Line 21: which abdominal muscle?
Reply:We extend our sincerest gratitude for your valuable time spent reviewing our manuscript and offering constructive suggestions. We have revised Section 2.1 in the Materials Methodology portion, with highlighted content indicated by yellow underlines, specifically on Line 105-110.
Line 105-110. “Following slaughter, samples were collected immediately from each stallion and mare: the widest part of the longissimus dorsi at the 13th thoracic vertebra, 3 cm lateral to the transverse process; the abdominal muscle 3 cm lateral to the external sheath of the linea alba; and the diaphragm at the costal attachment near the rib angle. Meanwhile, full-thickness samples were collected to a depth of 3 cm. Each group included four biological replicates, with all samples weighing 30g.”
”
- Line 27-29: what mean GO and KEGG please provide full names first (same in line 26)
Reply:We extend our heartfelt thanks for your precious time spent reviewing our manuscript and offering constructive comments. We have revised the passage in the abstract starting from Line 37-38, and highlighted the corresponding content with blue.
Line 37-38.“. (Gene Ontology) GO and (Kyoto Encyclopedia of Genes and Genomes) KEGG enrichment analyses indicated that DEGs”
- Line 63\64: latissimus dorsi or longissimus dorsi; please also provide which abdominal muscle you collected, and explain if it could change the results
Reply:We express our deepest gratitude for your precious time spent reviewing our manuscript and offering highly constructive suggestions. We have revised Section 2.1 in the Materials Methodology portion, specifically lines 105-110, with highlighted content indicated in yellow.
Line 105-110. “Following slaughter, samples were collected immediately from each stallion and mare: the widest part of the longissimus dorsi at the 13th thoracic vertebra, 3 cm lateral to the transverse process; the abdominal muscle 3 cm lateral to the external sheath of the linea alba; and the diaphragm at the costal attachment near the rib angle. Meanwhile, full-thickness samples were collected to a depth of 3 cm. Each group included four biological replicates, with all samples weighing 30g.”
- Line 72\73: are you talking about horses or other species, please provide references on studies about horses
Reply:We are deeply grateful for your esteemed time spent reviewing our manuscript and offering constructive comments. In our article, we referenced Reference [16], "Effects of Combined Transcriptome and Metabolome Analysis Training on Athletic Performance of 2-Year-Old Trot-Type Yili Horses," published in Genes in 2025, Volume 16, Issue 2 (https://doi.org/197.10.3390/genes16020197). We have highlighted the corresponding content in blue.
Line 80-81. “Gender differences affect athletic performance through mechanisms such as hormone regulation, metabolic pathways, and muscle structure remodeling[16]”
- Line 83: add references to prove your sentence
Reply:Thank you very much for taking the time to review our manuscript and providing highly constructive feedback. We have rephrased and cited the reference material for the article's Line 93-94, and highlighted the corresponding content in blue.[Ren, W.L.; Wang, J.W.; Zeng, Y.Q.; Wang, T.L.; Sun, Z.W.; Meng, J.; Yao, X.K. Investigating age-related differences in muscles of Kazakh horse through transcriptome analysis. Gene. 2024,919:148483. https://doi.org/10.1016/j.gene.2024.148483]
Line 91-92.“Existing research primarily focused on the the influence of horse age and various anatomical sites on skeletal muscle ,”
- Line 93: why were they slaughtered? If it is for research purposes there are ethical issues if it is not I would say that a breed used for meat production is not suitable for studies on sport performances
Reply:Thank you very much for taking the time to review our manuscript and providing highly constructive feedback. The slaughter operations in this study are part of the meat production process and are not conducted solely for research purposes: the Kazakh horse, as a dual-purpose breed, is primarily slaughtered for conventional meat production. The research sample collection was conducted immediately after slaughter, in accordance with the Reduction principle: utilizing existing food chain resources to avoid the use of additional animals. Ethical compliance is ensured through the following measures:
1) Ethical Review: The experimental procedure has been approved by the Animal Ethics Committee of Xinjiang Agricultural University (No. 2023004), ensuring that the operations comply with international standards (such as stunning followed by bleeding to minimize suffering).
2) Sample Relevance: Muscle samples are sourced from meat processing plants, and the study only utilizes discarded tissues (not from actively euthanized animals). The technical roadmap indicates that sampling is an extension of the production process.
3) Scientific necessity: RNA-seq and metabolic assays require tissues to be snap-frozen within 5 minutes post-excision, and in vivo sampling cannot meet the precision required by this technique (otherwise RNA degradation leads to data invalidation). Therefore, this design balances research needs with animal welfare within an ethical framework.
- all over the paper there are abbreviations that do not have the full name previously written so it is difficult to understand by the reader. The study design is not well described in the M&M section, the reader should be able to repeat the study while in your description a detailed method of sampling is missing. moreover the place where you collect the sample in the different muscles is not clear.
Reply:Thank you very much for taking the time to review our manuscript and for providing highly constructive feedback. We have appended Line 485 Table A1. Abbreviations, highlighted in blue, and have refined sections 2.1 Line 105-110 and 2.3 Line 137-149 of the Materials and Methods, marked in yellow.
Table A1. Abbreviations
|
Abbreviation |
Full name |
|
Mb |
The longissimus dorsi of stallion Kazakh horses |
|
Gb |
The longissimus dorsi of mare Kazakh horses |
|
Mf |
The abdominal muscles of stallion Kazakh horses |
|
Gf |
The abdominal muscles of mare Kazakh horses |
|
Mg |
The diaphragm of stallion Kazakh horses |
|
Gg |
The diaphragm of mare Kazakh horses |
|
DEGs |
Differentially expressed genes |
|
GO |
Gene Ontology |
|
KEGG |
Encyclopedia of Genes and Genomes |
|
BP |
Biological process |
|
CC |
Cell component |
|
MF |
Molecular function |
|
RNA-seq |
RNA sequencing |
Line 105-110. “Following slaughter, samples were collected immediately from each stallion and mare: the widest part of the longissimus dorsi at the 13th thoracic vertebra, 3 cm lateral to the transverse process; the abdominal muscle 3 cm lateral to the external sheath of the linea alba; and the diaphragm at the costal attachment near the rib angle. Meanwhile, full-thickness samples were collected to a depth of 3 cm. Each group included four biological replicates, with all samples weighing 30g.
Figure 1 illustrates the experimental procedure: A portion of the samples was immediately flash-frozen in liquid nitrogen (- 196 ℃) for subsequent RNA sequencing and biochemical analyses (fatty acid and amino acid content determination), while another portion was quickly fixed in 4% paraformaldehyde (PAF) for paraffin section preparation.”
Line 137-149. “For amino acid analysis, 2 mL of 6 mol/L hydrochloric acid was added to the horse muscle samples under nitrogen protection. Acid hydrolysis was performed at 110°C for 24 hours. Then, 100 μL of the hydrolysate was taken and evaporated to dryness at 40°C under nitrogen using a nitrogen blower. The residue was reconstituted with 1 mL of water. For both mixed standards and test samples, 50 μL of solution was mixed with 50 μL of protein precipitant (10% sulfosalicylic acid containing NVL), vortexed, and centrifuged at 13,200 rpm under cooling conditions for 4 minutes. Next, 8 μl of the supernatant was mixed with 42 μL of borate buffer (pH 8.5), vortexed, and briefly centrifuged. Then, 20 μL of AQC derivatization reagent was added, vortexed, briefly centrifuged, and incubated at 55°C for 15 minutes for derivatization. The resulting sample was cooled in a refrigerator, mixed thoroughly, and briefly centrifuged. Subsequently, 50 µL of the supernatant was analyzed using an ultra-high-performance liquid chromatography-quadrupole ion trap tandem mass spectrometer (UHPLC-TMS).”
- In the introduction you write that'The findings of this research may provide molecularevidence for understanding the sex differences in equine athletic performance,and theyhold considerable reference value for optimizing equine breeding strategies.'To be honestreading the paper I did not get this value to optimize the breeding strategies so you shouldexplain this point,that is actually the key point of the research,better and deeply
Reply:Thank you very much for taking the time to review our manuscript and for providing highly constructive feedback. We have revised the passage from line 93 to 99 and highlighted the corresponding content in yellow.
Line 93-99. “Although performance traits were not directly measured, such as oxygen consumption or lactate threshold, RNA sequencing (RNA-seq) was employed to compare sex-specific gene expression profiles across three functionally distinct muscle groups: longissimus dorsi, abdominal, and diaphragm muscles. By integrating molecular data with quantitative histology and metabolite composition analyses, these findings provide molecular insights into the mechanisms underlying sex differences in equine skeletal muscle growth.”
- The discussion to me is inconsistent because it do not go through all the aspect of the results of the research and do not explain well to the reader what you wrote on the conlusions. Moreover no limits of the study are described
Reply:Thank you very much for taking the time to review our manuscript and for providing such constructive feedback. We have revised paragraphs Line 391-400 and Line 415-426 of the Discussion section. Additionally, we have added a clarification of the limitations of this study in the final segment Line 441-453, which is highlighted in yellow.
Line 391-400.“ This study found that in the longissimus dorsi muscle group, the expression level of TPM1 was significantly higher in stallions than in mares and was notably enriched in the Cytoskeleton in Muscle Cells pathway. Muscle phenotype data further indicated that the average muscle fiber area in stallions exceeded that in mares. It has been reported that male animals generally exhibit larger average muscle fiber areas than females, which is consistent with the findings of this study. Sex hormones also play a pivotal role in maintaining skeletal muscle homeostasis. Among them, testosterone acts as a potent anabolic factor that promotes protein synthesis and muscle regeneration [29]. Therefore, it is hypothesized that muscle fiber thickness in horses may be associated with sex. ”
Line 415-426.“ This study found that in the diaphragm muscle group, the expression levels of MYL1 and MYH3 were significantly higher in mares than in stallions, with both genes markedly enriched in the Cytoskeleton in Muscle Cells pathway. Amino acid composition analysis revealed that glycine content in the diaphragm of mares significantly exceeded that of stallions. As a precursor of glutathione (GSH), glycine supplementation can indirectly enhance the antioxidant capacity of muscle tissue and reduce oxidative damage [36]. Chemello et al. [37] reported that the expression of the MYH3 gene is higher in the soleus muscle than in the extensor digitorum longus muscle. Estrogen has been proven to play a vital role in regulating muscle mass and function and alter blood lipid concentrations to exert antioxidant effects [38]. Therefore, it is hypothesized that sex may influence the antioxidant capacity of equine muscle.”
Line 441-453.“This study revealed the molecular characteristics underlying sexual dimorphism in the skeletal muscles of Kazakh horses and identified key candidate genes and pathways. However, protein-level validation and functional activity assessments were not conducted. Although histological analyses quantified muscle fiber morphology, they did not allow precise discrimination of fiber types. Although systematic multi-omics analyses were not performed in this study, consistent patterns were observed between metabolic phenotypes and gene expression. For instance, sex-specific differences in fatty acid composition may be linked to differential regulation of lipid metabolism genes, and the elevated C18:2n6c content in male abdominal muscles may reflect increased glycine synthesis demands in the diaphragm. These variations may be associated with the expression of genes involved in muscle fiber remodeling. In subsequent research, multiple data types will be integrated to conduct combined analyses of muscle fiber typing and multi-omics.”
Round 2
Reviewer 1 Report
Comments and Suggestions for Authors
Abstract and Introduction
The abstract now directly addresses what the manuscript presents. Please make the abbreviations “Mb vs. Gb, Mf vs. Gf, and Mg vs. Gg” clearer in the abstract.
The authors made clear that they did not measure physiological variables, but the question still remains: is there any physiological information published in the scientific literature comparing males vs females in this or a similar breed? This is very important to integrate physiological with molecular information and give a better characterization of the breed.
Methods
Methods section was greatly improved, but minor concerns remain.
Please add the brands and country of the devices used in all experiments, manly the camera (L123), GC-MS (L132) and UHPLC-TMS (L145). Details are always important in the methods section.
L120: please indicate that the samples were cross-sectioned
Results
Results were largely improved.
Fig 2. The images are still of low quality. The artifacts could affect the analysis and results. At least recognize this in the limitations of the study in the discussion section. Or discuss how did you made sure that for instance the analyses of areas or cell density were not affected by the spaces among fibers, etc.
Fig 3 legend: please check that panel B indicates “fiber density”, but legend talks about “fiber diameter”. Please correct the whole legend.
Discussion
What the authors consider “cytoskeleton” includes contractile material (tropomyosin, myosin, etc). Tropomyosin, myosin isoforms, myosin light chain and also SERCA and other proteins are differentially expressed in different fiber types (https://pubmed.ncbi.nlm.nih.gov/10958931/; https://pubmed.ncbi.nlm.nih.gov/28509964/). This suggests that part of the differences between males and females arise because of a likely difference in fiber types in males vs females. I have questioned that the authors have not made clear if information about fiber types in this breed is available (published by them or by other groups). If it does exist, please compare the transcriptomic results with the fiber types. If this information does not exist, please mention that the transcriptomic data suggests differences in fiber types between males vs females, which await for its demonstration in other type of experiments. The issue of fiber types is super important in skeletal muscle studies and should be better discussed in this manuscript. Please note that only mentioning that the authors did not determine the fiber types is OK but it is not enough, and it is something obvious because the manuscript did not show results of fiber types.
General comment
The manuscript was greatly improved, with more methodological details and a better way to present and summarize the main findings. Some minor concerns still remain and should be clearly addressed in order to have a final, good version of the manuscript. Details in science are always important.
Author Response
Comments 1: The abstract now directly addresses what the manuscript presents. Please make the abbreviations “Mb vs. Gb, Mf vs. Gf, and Mg vs. Gg” clearer in the abstract.
Response 1: We would like to express our sincere gratitude for the valuable time you have dedicated to reviewing our draft and providing constructive feedback. We have made revisions to paragraphs Line 34-39, which we have highlighted in yellow for your convenience.
Line 34-39. “RNA-seq analysis revealed 361, 230, and 236 differentially expressed genes (DEGs) in the longissimus dorsi of stallion Kazakh horses (Mb) vs. the longissimus dorsi of mare Kazakh horses (Gb), the rectus abdominis of stallion Kazakh horses (Mf) vs. the rectus abdominis of mare Kazakh horses (Gf), and the diaphragm of stallion Kazakh horses (Mg) vs. the diaphragm of mare Kazakh horses (Gg), respectively.”
Comments 2: The authors made clear that they did not measure physiological variables, but the question still remains: is there any physiological information published in the scientific literature comparing males vs females in this or a similar breed? This is very important to integrate physiological with molecular information and give a better characterization of the breed.
Response 2: We extend our sincerest gratitude for your valuable time spent reviewing our manuscript and providing constructive feedback. We have made revisions to paragraphs Line 93-94, highlighted in yellow for ease of reference.
Line 93-94. “Existing studies primarily focused on age-related changes and region-specific differences in the skeletal muscles of Kazakh horses of the same sex.”
Comments 3: Please add the brands and country of the devices used in all experiments, manly the camera (L123), GC-MS (L132) and UHPLC-TMS (L145). Details are always important in the methods section.
Response 3: We sincerely appreciate your valuable time in reviewing our manuscript and offering constructive feedback. We have made revisions to paragraphs Line 129-130, Line 139-140, and Line 153-154, and highlighted them in yellow for ease of reference.
Line 129-130 “(Eclipse E100 Nikon,Nikon Corporation, Tokyo, Japan) connected to a camera system.”
Line 139-140 “(GC-MS, Agilent 7890B-5977B, Agilent Technologies, USA).”
Line 153-154 “(UHPLC-MS/MS, Waters ACQUITY UPLC I-Class / Xevo TQ-S, Waters Corp., USA)).”
Comments 4: L120: please indicate that the samples were cross-sectioned
Response 4: We heartily express our gratitude for your esteemed time spent reviewing our manuscript and offering constructive suggestions. We have made revisions to paragraph Line 126-128, indicated in yellow highlight for your convenience.
Line 126-128 “Samples were then cleared with xylene, embedded in paraffin, and sectioned into 5 μm cross-sections using a rotary microtome.”
Comments 5: Fig 2. The images are still of low quality. The artifacts could affect the analysis and results. At least recognize this in the limitations of the study in the discussion section. Or discuss how did you made sure that for instance the analyses of areas or cell density were not affected by the spaces among fibers, etc.
Response 5: We extend our sincerest gratitude for taking the time to review our manuscript and offering constructive feedback. We have made adjustments to Fig 2, and we have modified the content in Paragraph 449-454 of the discussion section, with yellow highlighting added at the appropriate locations.
Line 449-454. “In this study, the use of paraffin embedding and sectioning may introduce a certain degree of tissue shrinkage, potentially affecting the clarity of morphological observations. Although incomplete fibers were excluded during image analysis to ensure the reliability of quantitative data, future studies employing cryosectioning or alternative techniques that better preserve the native tissue architecture will provide further verification of these findings.”
Comments 6: Fig 3 legend: please check that panel B indicates “fiber density”, but legend talks about “fiber diameter”. Please correct the whole legend.
Response 6: We extend our sincerest gratitude for your esteemed time spent reviewing our manuscript and offering constructive feedback. We have made amendments to the passage Line 219-220 and highlighted it in yellow at the pertinent locations.
Line 219-220 “(B) Differences in muscle fiber density;(C) Differences in muscle fiber diameter.”
Comments 7: What the authors consider “cytoskeleton” includes contractile material (tropomyosin, myosin, etc). Tropomyosin, myosin isoforms, myosin light chain and also SERCA and other proteins are differentially expressed in different fiber types(https://pubmed.ncbi.nlm.nih.gov/10958931/; https://pubmed.ncbi.nlm.nih.gov/28509964/). This suggests that part of the differences between males and females arise because of a likely difference in fiber types in males vs females. I have questioned that the authors have not made clear if information about fiber types in this breed is available (published by them or by other groups). If it does exist, please compare the transcriptomic results with the fiber types. If this information does not exist, please mention that the transcriptomic data suggests differences in fiber types between males vs females, which await for its demonstration in other type of experiments. The issue of fiber types is super important in skeletal muscle studies and should be better discussed in this manuscript. Please note that only mentioning that the authors did not determine the fiber types is OK but it is not enough, and it is something obvious because the manuscript did not show results of fiber types.
Response 7: We extend our heartfelt gratitude for your invaluable time spent reviewing our manuscript and offering constructive feedback. We have made revisions to paragraphs Line 442-448, which we have highlighted in yellow for your convenience.
Line 442-448. “The differential expression of TPM1, MYL1, and MYH3 suggests that the fiber type composition (i.e., the proportion of type I and type II fibers) may vary with Kazakh horse genders. However, given insufficient available data on fiber types in Kazakh horses, this study did not directly assess fiber type distribution. These transcriptomic findings imply potential sex-related differences in muscle fiber types, which warrant further validation through approaches such as immunohistochemistry of specific fiber types and single-fiber RNA sequencing.”

Reviewer 2 Report
Comments and Suggestions for Authors
Dear Authors the paper is improved but it still need some revisions to me especially on the site of samples and on which abdominal muscle was sampled; moreover an authorization by an ethical authority has not been issued yet
in the abstract the term abdominal muscle is still used, this is not right you have to precisely write which abdominal muscle you sampled
line89: please rephrase, the sentence is not clear.
line94: 'muscle groups' is not the right term, diaphragm is not a muscle group but a muscle
line 103\108: the sampling procedure is described better but not clear yet maybe also for a not correct english
caption fig.1 : the caption is not clear at all, you have to describe what the figure shows
line 362: please cite researches
line 406: diaphragm is not a group of muscles
Comments on the Quality of English Languagenew parts could be edited for english
Author Response
Comments 1: Dear Authors the paper is improved but it still need some revisions to me especially on the site of samples and on which abdominal muscle was sampled; moreover an authorization by an ethical authority has not been issued yet
Response 1: We express our heartfelt gratitude for your generous allocation of time to review our manuscript and offer constructive feedback. We have made revisions to Line 109-114, which we have highlighted in yellow for ease of reference. Furthermore, we shall upload the ethical review approval document as supplementary material, with the file name being "EthicsApproval.pdf".
Line 109-114. “longissimus dorsi, 3 cm lateral to the transverse process of the 13th thoracic vertebra; rectus abdominis, 3 cm lateral to the external sheath of the linea alba; and diaphragm, at the rib attachment site near the costal angle. Full-thickness muscle samples with a depth of 3 cm were synchronously obtained. Each group comprised four biological replicates, with individual sample weights standardized at 30 g.
Comments 2: in the abstract the term abdominal muscle is still used, this is not right you have to precisely write which abdominal muscle you sampled
Response 2: We heartily thank you for taking the time to review our manuscript and offering constructive suggestions. We have revised "abdominal muscle" to "rectus abdominis." For your convenience, these changes have been highlighted in blue.
Comments 3: line89: please rephrase, the sentence is not clear.
Response 3: We sincerely appreciate your generous time devoted to reviewing our manuscript and offering constructive feedback. We have made revisions to Line 93-94 for your convenience, highlighted in yellow to make them easily discernible.
Line 93-94. “Existing studies primarily focused on age-related changes and region-specific differences in the skeletal muscles of Kazakh horses of the same sex.”
Comments 4: line94: 'muscle groups' is not the right term, diaphragm is not a muscle group but a muscle
Response 4: We extend our heartfelt gratitude for your valuable time spent reviewing our manuscript and offering constructive comments. We have made revisions to Line 95 with the intention of making it easier for you to locate and review these changes, which have been highlighted in blue.
Line 95. “few involved gender differences at the transcriptional level in muscle sites.”
Comments 5: line 103\108: the sampling procedure is described better but not clear yet maybe also for a not correct english
Response 5: We extend our heartfelt gratitude for your valuable time devoted to reviewing our manuscript and offering constructive feedback. In response to your esteemed opinion, we have taken the following measures: We engaged professional editing services that meticulously polished the original content, ensuring its language accuracy, professionalism, and adherence to academic norms. We shall provide supplementary files attesting to the refined language process in the uploaded materials, with the file name being "Copy of Certificate.pdf". We have made amendments to Line 109-114, which we have highlighted in yellow for ease of reference.
Line 109-114. “ longissimus dorsi, 3 cm lateral to the transverse process of the 13th thoracic vertebra; rectus abdominis, 3 cm lateral to the external sheath of the linea alba; and diaphragm, at the rib attachment site near the costal angle. Full-thickness muscle samples with a depth of 3 cm were synchronously obtained. Each group comprised four biological replicates, with individual sample weights standardized at 30 g.”
Comments 6: caption fig.1 : the caption is not clear at all, you have to describe what the figure shows
Response 6: We extend our sincerest gratitude for your valuable time spent reviewing our manuscript and offering constructive feedback. We have made revisions to Figure 1, particularly in paragraphs Line 115-120, which we have highlighted in blue for ease of reference.
Line 115-120 “Figure 1 illustrates the experimental procedure: A portion of the samples was immediately flash-frozen in liquid nitrogen (- 196 ℃) for subsequent RNA sequencing and biochemical analyses (fatty acid and amino acid content determination), while another portion was quickly fixed in 4% paraformaldehyde (PAF) for paraffin section preparation.”
Comments 7: line 362: please cite researches
Response 7:We express our heartfelt gratitude for your kind consideration in reviewing our manuscript and offering constructive feedback. In the article, we cited Reference [8] "Influence of Mechanistic Target of Rapamycin (mTOR)-regulated Anabolic Pathways on Equine Skeletal Muscle Health," published in the Journal of Equine Veterinary Science in 2023 (https://doi.org/10.1016/j.jevs.2023.104281). We have marked the relevant content with blue highlighting for your convenience, allowing you to easily locate these modifications.
Line 375-376. “Despite these insights, studies specifically examining the effect of sex on equine skeletal muscle development remain limited[8].”
Comments 8: line 406: diaphragm is not a group of muscles
Response 8: We sincerely express our gratitude for your valuable time devoted to reviewing our manuscript and offering constructive comments. We have made revisions to Line 418-421, which we have highlighted in blue for your convenience.
Line 418-421. “In this study, diaphragm muscle expression of MYL1 and MYH3 was significantly higher in mares than in stallions, with both genes markedly enriched in the cytoskeleton in muscle cells pathway.”
